# Improved regional scale groundwater representation by the coupling of the mesoscale Hydrologic Model (mHM v5.7) to the groundwater model OpenGeoSys (OGS)

Miao Jing[1], Falk Heße[1], Rohini Kumar[1], Wenqing Wang[2], Thomas Fischer[2], Marc Walther[2,3], Matthias Zink[1], Alraune Zech[1], Luis Samaniego[1], Olaf Kolditz[2,4], and Sabine Attinger[1,5]

[1]Department of Computational Hydrosystems, UFZ – Helmholtz Centre for Environmental Research, Permoserstr. 15, 04318 Leipzig, Germany
[2]Department of Environmental Informatics, UFZ – Helmholtz Centre for Environmental Research, Permoserstr. 15, 04318 Leipzig, Germany
[3]Institute of Groundwater Management, Technische Universität Dresden, Bergstr. 66, 01069 Dresden, Germany
[4]Applied Environmental Systems Analysis, Technische Universität Dresden, Dresden, Germany
[5]Institute of Earth and Environmental Sciences, University of Potsdam, Karl-Liebknecht-Str. 24–25, 14476 Potsdam, Germany

**Abstract.**

Most large-scale hydrologic models fall short in reproducing groundwater head dynamics and simulating transport process due to their over-simplified representation of groundwater flow. In this study, we aim to extend the applicability of the mesoscale Hydrologic Model (mHM v5.7) to subsurface hydrology by coupling it with the porous media simulator OpenGeoSys (OGS). The two models are one-way coupled through model interfaces GIS2FEM and RIV2FEM, by which the grid-based fluxes of groundwater recharge and the river-groundwater exchange generated by mHM, are converted to fixed-flux boundary conditions of the groundwater model OGS. Specifically, the grid-based vertical reservoirs in mHM are completely preserved for the estimation of land-surface fluxes, while OGS acts as a plug-in to the original mHM modeling framework for groundwater flow and transport modeling. The applicability of the coupled model (mHM#OGS v1.0) is evaluated by a case study in the central European meso-scale river basin - Nägelstedt. Different time steps, i.e. daily in mHM and monthly in OGS, are used to account for fast surface flow and slow groundwater flow. Model calibration is conducted following a two-step procedure using discharge for mHM and long-term mean of groundwater head measurements for OGS, respectively. Based on the model summary statistics, namely the Nash–Sutcliffe model efficiency (NSE), the mean absolute error (MAE), and the inter-quartile range error (QRE), the coupled model is able to satisfactorily represent the dynamics of discharge and groundwater heads at several locations across the study basin. Our exemplary calculations show that the one-way coupled model can take advantage of the spatially explicit modeling capabilities of surface and groundwater hydrologic models, and provide an adequate representation of the spatio-temporal behaviors of groundwater storage and heads, thus making it a valuable tool for addressing water resources and management problems.

# 1 Introduction

Large scale hydrologic models had been primarily developed to predict river discharge. To that end, these models typically use simplified representation of underlying hydrological processes, usually bucket-type expressions to describe water storage and flow inside the catchment (Refsgaard and Storm, 1995; Wood et al., 1997; Koren et al., 2004; Samaniego et al., 2010; Niu et al., 2011). Water is transmitted between different vertical and horizontal buckets by means of an infiltration-runoff partitioning algorithm, which can be normally expressed as a function of the water storage (Hrachowitz and Clark, 2017). Model parameters in these types of models are usually obtained via calibration to match the observed dynamics of streamflow time series. As a result, these types of conceptual models are generally good at predicting streamflow dynamics. However, all (bucket-type) hydrologic models simplify water flow processes by ignoring lateral flow, especially at a large scale. Thus, such models inevitably fall short of characterizing subsurface groundwater dynamics, where such lateral flows are dominant. In addition, estimates of groundwater storage and heads are particularly error prone due to the low sensitivity of groundwater (storage) to river flows.

The groundwater representation in these conceptual (bucket-type) hydrologic models is consequently not adequate in several aspects. First,these models aggregate the heterogeneity of typically stratified groundwater aquifers, and fall short in adequately representing groundwater heads and low flow conditions (Ameli et al., 2016; Hale and McDonnell, 2016). Second, these models often do not properly capture the dynamics of solute transport and retention at the catchment scale. For example, Van Meter et al. (2017) found that current nitrogen fluxes in rivers can be dominated by groundwater legacies. An over-simplified groundwater representation is inadequate for understanding travel-time distributions (TTDs) at a catchment scale and is therefore not capable of describing such legacy behavior (Botter et al., 2010; Benettin et al., 2015, 2017). Finally, a more accurate groundwater representation including lateral subsurface flow is needed to predict the response of groundwater to climate change (Scibek and Allen, 2006; Green et al., 2011; Ferguson et al., 2016).

Parallel to such conceptual surface hydrologic models, three dimensional (partial differential equation; PDE)-based sub-surface models have been developed, which allow for both steady-state and transient groundwater flow, accounting for the representation of sub-surface heterogeneity and varying degree of sources and sinks. Such model are good at tackling the aforementioned problems encountered in application of conceptual models. At this end, a variety of numerical codes/models are available such as: InHM (VanderKwaak and Loague, 2001; Smerdon et al., 2007), Parflow (Maxwell and Miller, 2005; Maxwell et al., 2015), OpenGeoSys (Delfs et al., 2012; Kolditz et al., 2012), tRIBS (Ivano et al., 2004), CATHY (Camporese et al., 2010), HydroGeoSphere (Therrien et al., 2010; Hwang et al., 2014), MODHMS (Panday and Huyakorn, 2004; Phi et al., 2013), GEOtop (Rigon et al., 2006), IRENE (Spanoudaki et al., 2009), CAST3M (Weill et al., 2009), PIHM (Kumar et al., 2009; Qu and Duffy, 2007), and PAWS (Shen and Phanikumar, 2010). PDE-based hydrologic models usually represent sub-surface flow by accounting for both saturated and unsaturated groundwater flows. The flow fields can be directly computed on the basis of spatial gradients of the modeled primary variable, e.g., the hydraulic head. The PDE-based models are flexible in coping with subsurface heterogeneity by means of proper characterization of aquifer system (e.g. stratification or geostatistical approach), and thus are able to reduce aggregation errors caused by geological heterogeneity (de Marsily et al., 2005; Cirpka

and Attinger, 2003; Zech et al., 2015). Furthermore, these models can explicitly compute flow pathlines and provide direct estimates of travel times of water and solute particles. These properties of PDE-based models provide a significant advantage over (bucket-type) conceptual models especially in complex real-world applications (Park et al., 2008; Engdahl and Maxwell, 2015; Danesh-Yazdi et al., 2018).

However, despite these advantages in modeling the deeper subsurface flows, PDE-based models are not without problems, in particular in capturing the near-surface flow dynamics i.e., in shallow portions of the subsurface. For example, the PDE-based models often encounter problems in the unsaturated zone for simulating the quick flow components, which are mainly dependent on subgrid heterogeneities of topographic variation as well as soil and land-cover characteristics (Paniconi and Putti, 2015). Using a complex PDE-based surface hydrologic model to simulate near-surface processes is possible in general, but it

does require a model implementation at a fine spatial resolution to resolve the sub-grid features (e.g., root-water uptake) and include a tremendous number of uncertain model parameters. Furthermore these models have dramatically increased numerical complexity and computation time, and thus calibrating these models is a cumbersome task, and doing this in a stochastic framework is computationally not feasible.

     To summarize these points, bucket-type hydrologic models, such as mHM (Samaniego et al., 2010; Kumar et al., 2013b), VIC

(Liang et al., 1994), and HBV (Lindström et al., 1997), are good at predicting water fluxes, such as discharge, but are highly conceptual and their model results are difficult to interpret with respect to certain processes (e.g., groundwater flows). The outputs of PDE-based hydrologic models, such as Parflow, CATHY, and HydroGeoSphere, are highly interpretable but show consistently worse performance than bucket-type models when predicting discharge dynamics (Gulden et al., 2007; Paniconi and Putti, 2015). The differing capabilities of these two types of models are the result of the different challenges that are posed

by the various compartments of the terrestrial water cycle. One of the main challenges in modeling surface and near-surface storage is process uncertainty. The process uncertainty is caused by the strong non-liearities of hydrological processes and the fine-scale variability of land-surface features. Thus it can be hardly solved by PDE-based models, but can be well handled by bucket-type models through the parameterization process. In the deeper subsurface storage, the temporal and spatial scale of groundwater process is significantly larger than the shallow storage, and the flow is governed by linear PDE (Darcy's law), thus

makes the PDE-based model standard at a large scale (Dagan, 2012). Meanwhile, the data uncertainty becomes more significant in the deep subsurface storage in comparison to shallow storage due to the spatially sparse hydrogeological data. Moreover, a recent study reveals the strong spatial and temporal heterogeneity of processes and properties at the surface/groundwater (SW/GW) interface, and underlines the importance of quantifying variability across several scales at the SW/GW interface and its significance to water resources management (McLachlan et al., 2017). It therefore stands to reason, that the use of a hybrid

of both these model frameworks is a good choice for a joint representation of surface and subsurface water storages and fluxes. Therefore, the coupling between the bucket-type land-surface model and the subsurface saturated/unsaturated groundwater model is highly relevant. Several well-tested coupled models have been developed in recent years, including ParFlow-CLM (Maxwell and Miller, 2005; Maxwell et al., 2015; Kurtz et al., 2016), GSFLOW (Markstrom et al., 2008; Hunt et al., 2013), PCR-GLOBWB-MOD (Sutanudjaja et al., 2011, 2014), and CP(v1.0) (Bisht et al., 2017).

In this study, we present a coupling between the conceptual mesoscale Hydrologic Model (mHM v5.7) with the PDE-based model OpenGeoSys (OGS) (Kolditz et al., 2012, 2016). The overall aim is to provide a proper representation of groundwater flows and storages and enabling the coupled model to provide reliable estimates of groundwater heads. mHM has demonstrated its pre-eminence in coping with near-surface process uncertainty while providing reliable representation of observed discharge behavior across range of scales and locations (Samaniego et al., 2010; Kumar et al., 2013b; Huang et al., 2017). On the other hand, OGS has demonstrated its capability of dealing with data uncertainty in aquifers (Sun et al., 2011; Walther et al., 2012; Selle et al., 2013). The general idea behind the coupling is to use the hydrological simualtion of with mHM, including the simplified linear groundwater storage, extract fluxes into- and out of groundwater from mHM, and use this as Neumann boundary condtion for the PDE-based groundwater model OGS. By doing so, we augment the predictive power of mHM to also predict hydraulic heads in groundwater. The one-way coupling approach considered here has a number of advantages. First, the one-way coupling can be regarded as a conservative approach, such that the parametrization process, which is one of the most significant features of mHM, remains fully intact. In particular, this means that the whole body of confidence in the predictive power of mHM, that has been build up over the years can be fully relied on. Second, using such a one-way coupling will allow users of mHM to simply extend currently established catchment models and enhance their abilities in the aforementioned way. Using a more sophisticated two-way coupling, would entail users having to rebuild their models almost entirely. Third, a one-way coupling allows for ready future expansion of the functionality of the coupled model, e.g., legacy of solutes in groundwater, should the need arises. Finally, one-way coupling takes less computational efforts and achieves better numerical stability than two-way coupling.

By coupling two well-tested model codes, we want to answer the following scientific questions: (1) Can spatially distributed groundwater heads and their dynamics be reasonably captured by expanding the capabilities of a surface hydrologic model, such as mHM at the regional-scale, while conserving its excellence in predicting discharge? (2) Can spatially resolved groundwater recharge estimates, provided by mHM, improve the prediction of head measurements of groundwater models such as OGS? To answer these questions, we applied the coupled model mHM#OGS v1.0 in a central German meso-scale catchment ($850 \, km^2$), and evaluated the model skills using measurements of streamflow and groundwater heads from several wells located in the study area. The coupled (surface) hydrologic and groundwater model (mHM#OGS v1.0) presented in this paper is our first attempt toward the development of a large-scale coupled modeling system with the aim to analyze the spatio-temporal variability of groundwater flow dynamics at a regional scale.

To answer these questions, the paper is structured as follows. In the next section, we describe the model concept, model structure, and the coupling scheme. In Section 3.1, the study area and model setup used for illustration in this study are comprehensively described. In Section 4, we present the simulation results of mHM#OGS v1.0 in a catchment in the application. In the Section 5, we discuss the model results as well as advantages and limitations of current modeling approach.

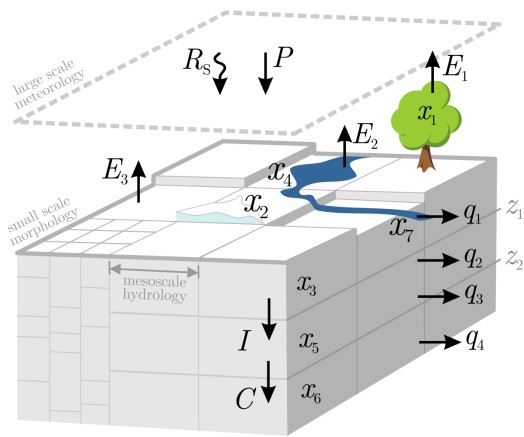

**Figure 1.** The concept of the mesoscale hydrologic model, mHM.

## 2  Model description

### 2.1  mesoscale Hydrologic Model (mHM)

The mesoscale Hydrologic Model (mHM, www.ufz.de/mhm) is a spatially explicit distributed hydrologic model that uses grid cells as a primary modeling unit, and accounts for the following processes: canopy interception, snow accumulation and melt-
ing, soil moisture dynamics, infiltration and surface runoff, evapotranspiration, subsurface storage and discharge generation, deep percolation, baseflow, discharge attenuation, as well as flood routing (Figure 1). The runoff generation applies a robust scheme which routes runoff in upstream cells along river networks using the Muskingum-Cunge algorithm. The model is driven by daily meteorological forcings (e.g., precipitation, temperature), and utilizes observable physical properties or signals of the basin (e.g., soil textural, vegetation, and geological properties) to infer the spatial variability of the required parameters. mHM
is an open-source project written in Fortran 2008. Parallel versions of mHM using OpenMP concepts are available.

A unique feature of mHM is the application of Multiscale Parameter Regionalization (MPR). The MPR method accounts for subgrid variability in physical characteristics of the catchment such as topography, soil and vegetation. The MPR methodology facilitates the flexibility of the model for hydrological simulations at various spatial scales (Samaniego et al., 2010; Kumar et al., 2013a, b; Rakovec et al., 2016a, b; Samaniego et al., 2017). mHM differentiates three levels to better represent the
spatial variability of state and input variables. The effective parameters at different spatial scales are dynamically linked by a physically-based upscaling scheme. A detailed description of MPR, as well as the formulations governing hydrological processes, are given by Samaniego et al. (2010) and Kumar et al. (2013b).

Below, we list the equations that describe near-surface processes in the deep soil and groundwater layers. The comprehensive system of equations of mHM can be found in Samaniego et al. (2010). Here, we only listed the equations needed for the
coupling. In the subsurface reservoir, which is the second vertical layer ($x_5$ in Figure 1), interflow is partitioned into fast

interflow ($q_2$) and slow interflow ($q_3$):

$$q_2(t) = max\{I(t) + x_5(t-1) - \beta_1(z_2 - z_1),\, 0\}\beta_2 \tag{1}$$

$$q_3(t) = \beta_3(x_5(t-1))^{\beta_4} \tag{2}$$

where $q_2(t)$ is fast interflow at time $t$ [LT$^{-1}$], $I$ is the infiltration capacity [L], $x_5$ is the water depth of water storage in the deep soil reservoir [L], $\beta_1$ is the maximum holding capacity of the deep soil reservoir [-], $z_i$ is depth of subsurface layer $i$ [L], $\beta_2$ is the fast-recession constant [T$^{-1}$], $q_3(t)$ is slow interflow at time $t$ [LT$^{-1}$], $\beta_3$ is the slow-recession constant, and $\beta_4$ is the exponent that quantifies the degree of non-linearity of the cell response.

The groundwater recharge is equivalent to the percolation to the groundwater reservoir (the third vertical layer, see $x_6$ in Figure 1). The groundwater recharge $C(t)$ can be expressed by

$$C(t) = \beta_5 x_5(t-1) \tag{3}$$

where $C(t)$ is the groundwater recharge in cell $i$ [LT$^{-1}$], and $\beta_5$ is the effective percolation rate coefficient [T$^{-1}$].

In the groundwater reservoir, baseflow is generated following a linear relationship between storage and runoff:

$$q_4(t) = \beta_6 x_6(t-1) \tag{4}$$

where $q_4(t)$ is the baseflow [LT$^{-1}$], $\beta_6$ is the baseflow recession rate coefficient [T$^{-1}$], and $x_6$ is depth of groundwater reservoir [L].

The runoff from upstream grid cells and the internal runoff in cell $i$ are routed into streams using the Muskingum algorithm:

$$Q_i^1(t) = Q_i^1(t-1) + c_1(Q_i^0(t-1) - Q_i^1(t-1)) + c_2(Q_i^0(t) - Q_i^0(t-1)) \tag{5}$$

with

$$Q_i^0(t) = Q_{i'}(t) + Q_{i'}^1(t) \tag{6}$$

$$c_1 = \frac{\Delta t}{\kappa(1-\xi) + \frac{\Delta t}{2}} \tag{7}$$

$$c_2 = \frac{\frac{\Delta t}{2} - \kappa\xi}{\kappa(1-\xi) + \frac{\Delta t}{2}} \tag{8}$$

where $Q_i^0$ and $Q_i^1$ denote the runoff entering and leaving the river reach located in cell $i$, respectively [LT$^{-1}$], $Q_{i'}$ is the contribution from the upstream cell $i'$ [LT$^{-1}$], $\kappa$ is the Muskingum travel time parameter [T], $\xi$ is the Muskingum attenuation parameter [-], $\Delta t$ is the time step-size [T], and $t$ is the time index for each $\Delta t$ interval.

## 2.2  OpenGeoSys (OGS)

OpenGeoSys (OGS) is an open-source project with the aim of developing robust numerical methods for the simulation of Thermo-Hydro-Mechanical-Chemical (THMC) processes in porous and fractured media. OGS is written in C++ with a focus

on the finite element analysis of coupled multi-field problems. Parallel versions of OGS based on both MPI and OpenMP concepts are available (Wang et al., 2009; Kolditz et al., 2012; Wang et al., 2017). To date, two OGS versions are available: OGS5 (https://github.com/ufz/ogs5) and OGS6 (https://github.com/ufz/ogs). In this study, the term "OpenGeoSys (OGS)" represents OGS5 if not stated otherwise.

OGS has been successfully applied in different fields, such as water resources management, hydrology, geothermal energy, energy storage, $CO_2$ storage, and waste disposal (Kolditz et al., 2012; Shao et al., 2013; Gräbe et al., 2013; Wang et al., 2017). In the field of hydrology / hydrogeology, OGS has been applied to regional groundwater flow and transport (Sun et al., 2011; Selle et al., 2013), contaminant hydrology (Beyer et al., 2006; Walther et al., 2014), reactive transport (Shao et al., 2009; He et al., 2015), and sea water intrusion (Walther et al., 2012), among others.

Saturated groundwater flow follows the continuity equation and Darcy's law:

$$S\frac{\partial \psi_p}{\partial t} = -\nabla \cdot \boldsymbol{q} + q_s \tag{9}$$

$$\boldsymbol{q} = -K_s \nabla(\psi_p + z) \tag{10}$$

where $S$ is specific storage coefficient in confined aquifers, and the specific yield in unconfined aquifers [1/L], $\psi_p$ is the pressure head in the porous medium [L], $t$ is time[T], $\boldsymbol{q}$ is the specific discharge or Darcy velocity [LT$^{-1}$], $q_s$ is the general source/sink

term [T$^{-1}$], $K_s$ is the saturated hydraulic conductivity tensor [LT$^{-1}$], and $z$ is the vertical coordinate (positive upward) [L].

The stream network is normally represented by a set of polylines in the geometry file of OGS. In the case of a 3-D model, a common way to set up the polyline system is to utilize the mapping tool embedded in OGS source codes, by which the shape file obtained from GIS software representing streams can be easily mapped onto the upper surface of OGS mesh and converted into a set of polylines. Each reach of the stream network can be represented by one polyline or several continuous polylines,

depending on the demand of the user. Each polyline consists of a set of continuous mesh nodes, upon which Dirichlet, Neumann or Robin boundary conditions can be applied.

## 2.3   Coupling mechanism

The coupled model mHM#OGS v1.0 is developed to simulate SW/GW flow in one or more catchments by simultaneously calculating flow across the land surface and within the groundwater. mHM#OGS v1.0 simulates flow within three hydrological

regions. The first region is limited by the upper boundary of the plant canopy and the lower boundary of the soil zone bottom. The second region includes open-channel water, such as streams. The third region is the water-saturated aquifer. mHM is used to simulate the processes in the first and second regions, while OGS is used to simulate the simulate groundwater flow for prescribed fluxes at all boundaries in the third region.

The coupling initiative aims to add additional predictive capability of groundwater heads, which is achieved by OGS, to the

existing capability of predicting discharges that is achieved by mHM. mHM is used to estimate step-wise and component-wise a water budget through model calibration against discharge. In contrast, OGS serves as a post-processor to obtain groundwater heads by using mHM simulated recharge and baseflow as driving forces. Two model interfaces, namely GIS2FEM and RIV2FEM, have been developed to link the two models by transferring recharge and baseflow from mHM to Neumann bound-

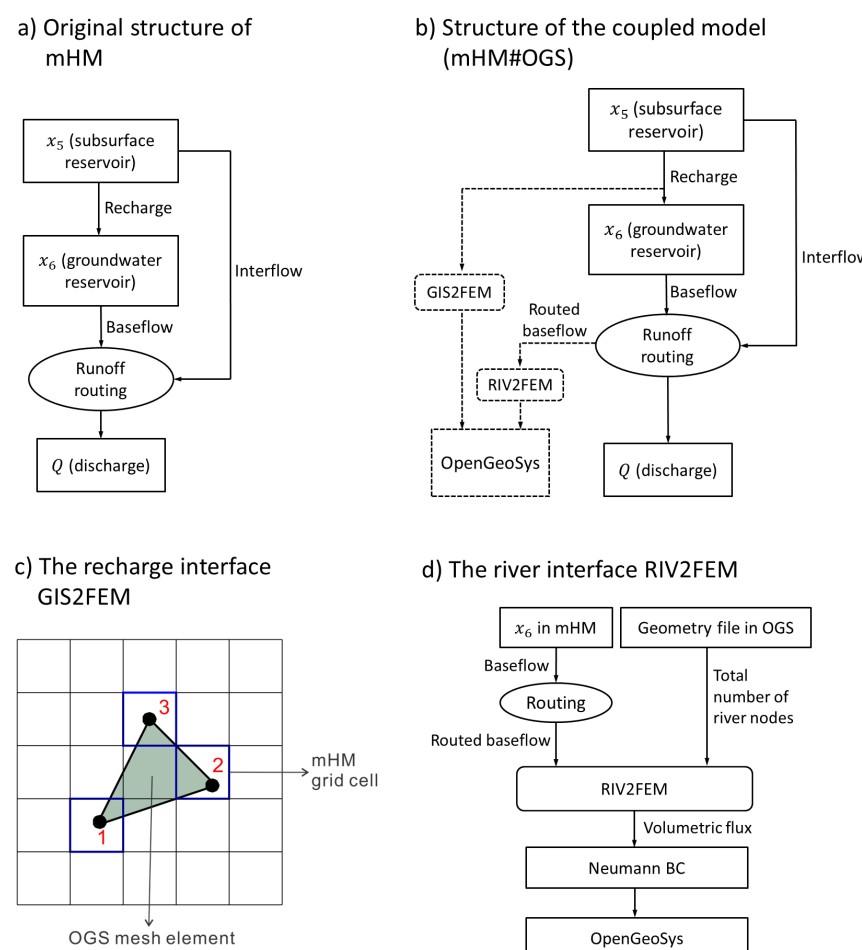

**Figure 2.** Schematic of the coupled model mHM#OGS v1.0. a) Original structure of the vertically layered reservoir of mHM. b) Structure of the coupled model (mHM#OGS v1.0). c) Illustration of data interpolation and transformation through the coupling interface GIS2FEM. d) Scheme of the river interface RIV2FEM. For the sake of simplicity, the figure only displays mHM layers relevant to this study and neglects the other mHM layers (i.e. $x_1$ - $x_4$). In Figure 2c, the grid-based mHM fluxes (e.g., recharge) are linearly interpolated to the top surface of the OGS mesh, and further transferred into volumetric values and directly assigned to the surface mesh nodes of the OGS grid.

ary conditions in OGS. The two models are executed separately and sequentially, typically with different temporal (e.g., daily in mHM and weekly or monthly in OGS) and spatial resolutions (e.g., rectangular, structured grids with coarse resolution in mHM and smaller, potentially unstructured grids with fine resolution in OGS). The original vertically layered reservoirs in mHM, namely the soil-zone reservoir and the subsurface reservoir are preserved, implying that all well-tested features of mHM (e.g., MPR, infiltration-runoff partitioning) are retained in the coupled model.

To illustrate the coupling mechanism in detail, we itemized the coupling workflow below.

1. mHM is run independently of OGS to calculate land surface fluxes including exchange fluxes of the groundwater storage.

   Using gridded meteorological forcings (precipitation, temperature, and potential evapotranspiration), the grid-based infiltration rates (e.g., groundwater recharge) and runoff components (e.g., interflow, baseflow) are estimated and saved as mHM output files. The original linear groundwater reservoir (depth $x_6$ in Figure 1) is used to estimate baseflow. Moreover, MPR is used in the calibration process such that subgrid variabilities can be validly calculated. The spatially distributed groundwater recharge and total routed baseflow are written into raster files for later use.

2. After the mHM run has finished, the step-wise routed baseflow estimated by mHM is transformed to distributed river discharges along streams and spatially distributed exchange rates between streams and groundwater needed in OGS.

   Most PDE-based models characterize river-groundwater interaction based on either first-order flow exchange or boundary condition switching (Paniconi and Putti, 2015). However, these approaches inevitably introduce extra parameters describing geometric, topographic, and hydraulic properties of the stream channel (e.g., river bed conductance, river bed and drain elevations, channel width). Unfortunately, these parameters are essentially unknown at a large scale due to the lack of data and the subgrid-scale variability of these parameters. Due to these limitations, we use an alternative approach which is based on the routed baseflow estimated by mHM.

   mHM and OGS conceptualize streams differently: streams in mHM are implicitly defined based on pre-processing of digital elevation model (DEM) data and a routing scheme, while OGS uses an explicit predefined river geometry. In OGS, each reach of the stream network is defined by a polyline in the OGS geometry file. To coordinate the two different approaches, we developed a model interface, RIV2FEM, to convert the routed baseflow estimated by mHM to Neumann boundary conditions assigned at stream nodes of the OGS mesh (Figure 2d). Via RIV2FEM, the routed baseflow estimated by mHM is transferred to the uniformly disaggregated groundwater discharges by distributing it uniformly along the predefined stream network in OGS (Figure 2d):

$$\overline{q}_4(t) = \frac{Q_4(t)}{\sum_{i=1}^N A_i} \tag{11}$$

   where $\overline{q}_4(t)$ denotes the normalized flux of disaggregated groundwater discharge at time $t$ [LT$^{-1}$], $Q_4(t)$ denotes the routed baseflow at the outlet of catchment at time $t$ [L$^3$T$^{-1}$], $A_i$ is the nodal area of the $i$th stream node [L$^2$], $N$ is the total number of stream nodes. The uniformly disaggregated groundwater discharges are then assigned to every stream node in OGS to serve as the Neumann boundary condition (Figure 2d). This approach significantly reduces the number of parameters, avoids the uncertainty caused by the unknown river properties, and is suitable for many real-world applications that suffer from scarce data. Moreover, as recharge and baseflow are directly taken from mHM, the mass conservation criterion is naturally satisfied in this approach.

3. The distributed groundwater recharge generated by mHM is fed to the coupling interface GIS2FEM, and then transferred to the upper surface boundary conditions of the OGS model.

   The coupling interface GIS2FEM is used to interpolate and transfer mHM grid-based recharge to OGS nodal recharge values. GIS2FEM interpolates the flux value to the top surface elements of the OGS mesh. The detailed workflow is:

- GIS2FEM reads the raster file generated by mHM and the mesh file of OGS.

- In the case of a 3-D mesh, GIS2FEM extracts the upper surface of the OGS mesh. For each of the nodes on this surface, GIS2FEM searches for the mHM grid cell that the node is located in, and assigns the recharge value of this grid cell to the corresponding node (marked as $C^m$).

- After all top surface elements have been processed, GIS2FEM undertakes the face integration calculation, by which the specific recharge $C^m$ [LT$^{-1}$] calculated by mHM is converted into volumetric recharge $C^{in}$ [L$^3$T$^{-1}$] and assigned to the corresponding OGS mesh nodes (Figure 2c). Specifically, the specific recharge $C$ in a certain element is calculated as:

$$C(\mathbf{x}) = \sum_{i=1}^{N} W_i(\mathbf{x})C_i^m,$$
(12)

where $\mathbf{x}$ is the spatial coordinate on the surface, $N$ is the total number of nodes in a surface element, $W_i$ is the weighting function of $i$th node of OGS surface element, $C_i^m$ is the specific recharge at $i$th node of OGS surface element (calculated by mHM) [LT$^{-1}$]. Then the volumetric recharge $C_i^{in}$ at $i$th node (here $i$ is the global node index) is calculated by the face integration calculation:

$$C_i^{in} = -\int_{\partial\Omega} W_i(\mathbf{x})C(\mathbf{x})d(\mathbf{x}),$$
(13)

where $C_i^{in}$ is the volumetric recharge of node $i$ [L$^3$T$^{-1}$], $\partial\Omega$ is the surface boundary of the FEM domain, $W_i$ is the weighting function of $i$th node.

4. After the mHM-generated recharge and baseflow have been transferred into boundary conditions at the upper surface of the OGS mesh, the groundwater model is run to simulate the groundwater flow and transport.

In this step, additional boundary conditions can be set up in OGS mesh on the basis of expert knowledge. The exclusive use of Neumann BC is not recommended and may lead to nonuniqueness of solutions. At least one specified head boundary should be set at the perimeter or internal nodes to constrain the model solution. The groundwater model simulates groundwater flow to obtain hydraulic heads in the example application. The groundwater model may also be used to compute travel times and solute transport within the groundwater domain, requiring additional boundary conditions; but this is not described in the present paper.

# 3 Example application

## 3.1 Study area and model setup

We use a meso-scale catchment (about 850 km$^2$) upstream of the Nägelstedt gauge located in central Germany to test our coupled model (Figure 3). The Nägelstedt catchment comprises the headwaters of the Unstrut River, a tributary of the River Saale. We selected this study area because many of the groundwater monitoring wells in the area are operated by the Thuringian

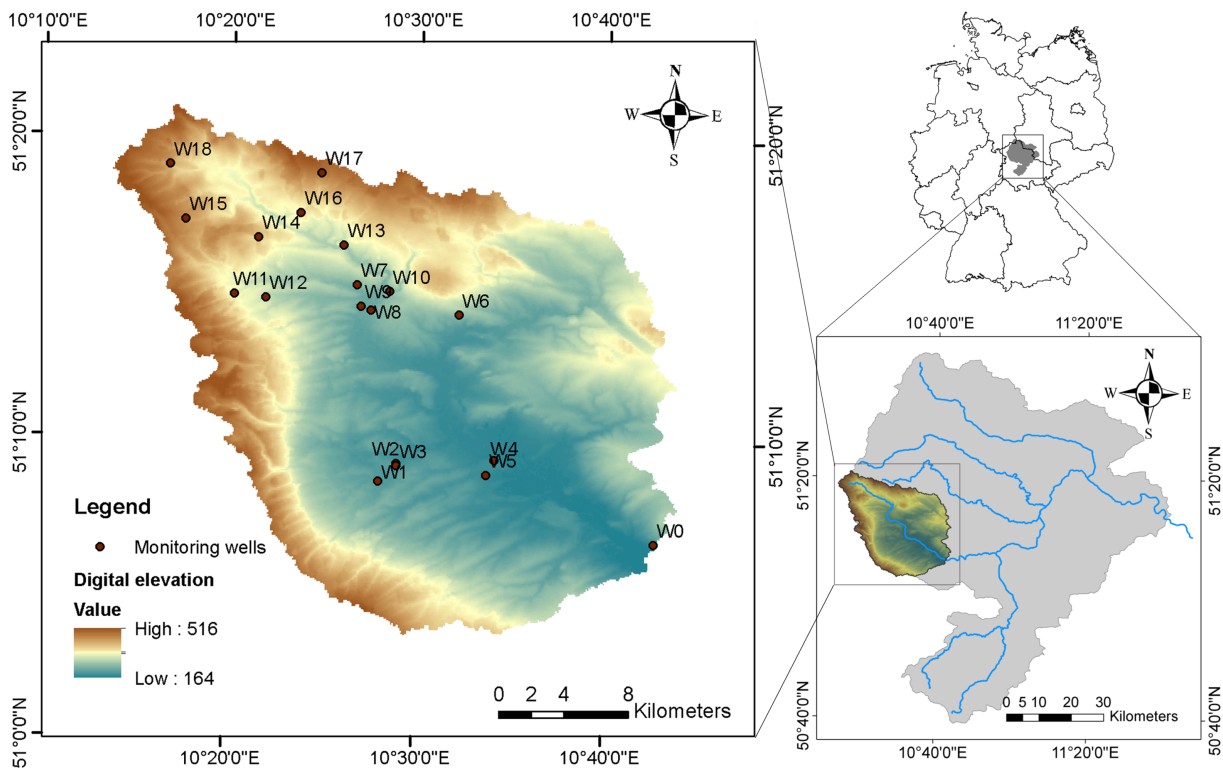

**Figure 3.** The Nägelstedt catchment used as the test catchment for this study. The left-hand map shows elevation and locations of monitoring wells used in this study. The lower right-hand map shows the relative location of Nägelstedt catchment in the Unstrut Basin. The upper right-hand map shows the location of the Unstrut Basin in Germany.

State Office for the Environment and Geology (TLUG) and the Collaborative Research Center AquaDiva (Küsel et al., 2016). The elevation within the catchment ranges between 164 m and 516 m, whereby the higher regions are in the west and south and belong to the forested hill chain of the Hainich (Figure 3). The Nägelstedt catchment is one of the most intensively used agricultural regions in Germany. In terms of drinking water supply, about 70% of the water requirement is satisfied by groundwater (Wechsung, 2005). About 17% of the land in this region is forested area, 78% is covered by crop and grassland, and 4% is urban and transport area. The mean annual precipitation in this area is about 660 mm.

In this study, mHM runs were executed for a time period of 35 years (from January 1, 1970 to December 30, 2004), with the period 1970 - 1974 being used for spin-up. OGS was run for the period from January 1, 1975 to December 30, 2005. mHM was run with a daily time step, while OGS was run with a monthly time step. The resolution of mHM grid cells is 500 m × 500 m. OGS uses a structured, hexahedral 3-D mesh, with a spatial resolution of 250 m × 250 m in the horizontal direction and 10 m in the vertical direction over the whole domain. The detailed input data and parameter-set to run both models are detailed in the following sections.

## 3.2 Meteorological and surface properties

We started the modeling by performing the daily simulation of mHM to calculate near-surface hydrological processes. The mHM model is forced by daily meteorological conditions, including distributed precipitation and atmospheric temperature. The spatial patterns of precipitation and atmospheric temperature were based on point measurements of precipitation and

atmospheric temperature at weather stations from the German Meteorological Service (DWD). The point data at weather stations were subsequently kriged onto a 4 km precipitation field, and then downscaled to mHM grid cells. Moreover, the potential ET was estimated based on the method from Hargreaves and Samani (1985). Other datasets used in mHM are the DEM data, which is the basis for deriving properties such as slope, river beds, and flow direction; soil and geological maps, and derived properties such as sand and clay contents, bulk density; CORINE land-cover information (in the years 1990, 2000

and 2005); and discharge data at the outlet of the catchment.

## 3.3 Aquifer properties

We used a stratigraphic model to explicitly represent the heterogeneous distribution of hydraulic properties (hydraulic conductivity, specific yield, and specific storage). The stratigraphic model is based on well log data and geophysical data obtained from the Thuringian State Office for the Environment and Geology (TLUG). We used the workflow developed by Fischer et al.

(2015) to convert the data format, by which the complex 3-D geological model was converted into the open-source VTK format file that can be directly read by OGS.

The major stratigraphic units in the study site are the Muschelkalk (Middle Triassic) and the Keuper (Upper Triassic). Younger Tertiary and Quaternary deposits are less important for the large-scale hydrogeology of the basin. The Keuper deposits mainly lie in the center of the Unstrut Basin and act as permeable shallow aquifers. In the Nägelstedt catchment, the Keuper

deposits are further subdivided into two geological sub-units: Middle Keuper (km) and Lower Keuper (ku) (see Figure 4). The Muschelkalk is marked by a prevailing marine environment and is subdivided into three sub-units the Upper Muschelkalk (mo), Middle Muschelkalk (mm, dolomites and residues of eroded salt layers) and Lower Muschelkalk (mu, limestones). According to previous geological surveys (Seidel, 2004), the sub-units of the Muschelkalk have varying hydraulic properties depending on their positions and depths. They are further divided into sub-units with higher permeabilities (mo1, mm1 and mu1) and sub-

units with lower permeabilities (mo2, mm2 and mu2) (Figure 4). The Upper Muschelkalk (mo) has been widely considered as a karstified formation. Recent research by Kohlhepp et al. (2017) has revealed that in the Hainich Critical Zone, the intense karstification and the conduit are limited at the base of the mo formation. Accordingly, we use the equivalent porous medium approach to characterize the Upper Muschelkalk. The uppermost layer with a depth of 10 m is set as a soil layer (Figure 4). A high-permeability alluvium layer is set along the mainstream and major tributaries to represent granite and stream deposits

(Figure 4).

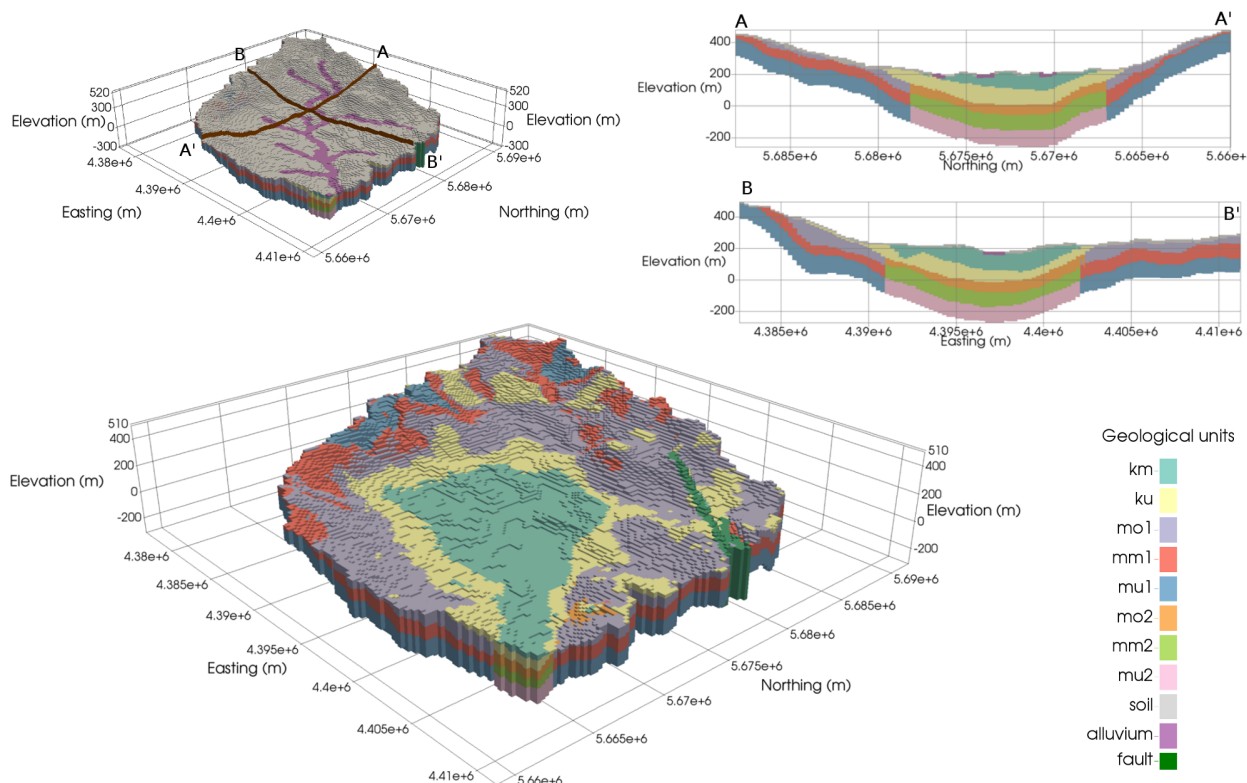

**Figure 4.** Three-dimensional and cross-sectional views of the hydrogeologic zonation in the Nägelstedt catchment. The upper left-hand figure highlights the distribution of alluvium and soil zones. The upper right-hand figure shows two vertical geological cross-sections. The lower map shows the detailed zonation of geological sub-units beneath the soil zone and alluvium.

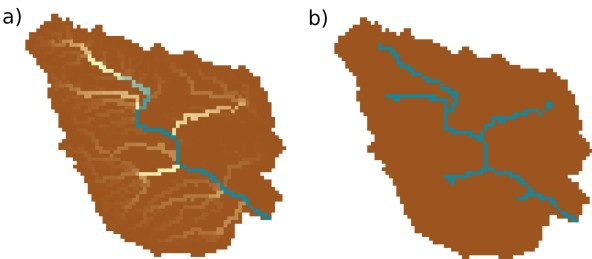

**Figure 5.** Illustration of the stream network used in this study. a) Original stream network based on the streamflow routing algorithm of mHM; b) Processed stream network that was used in this study. The small tributaries where the runoff rates are below the threshold (0.145 m$^3$/s) as shown in the left figure, have been removed to form the right figure.

### 3.4 Boundary conditions

Based on the steep topography along the watershed divides, groundwater is assumed to be naturally separated and unable to pass across the boundaries of the watershed. In general, no-flow boundaries are set at the outer perimeters surrounding the basin as well as at the lower aquitard, except for the northwestern and northeastern edges. On the basis of the measurements, a Dirichlet boundary condition is assumed at the northwestern and northeastern edges.

The stream network was delineated by processing a grid-based runoff raster file generated by mHM. The grid-based runoff was converted to a valid stream network compatible with OGS. The necessity of transferring the mHM runoff raster file to the OGS stream network has been elaborated in Section 2.3. Particularly in this case study, we removed the small intermittent tributaries by setting a threshold value of long-term averaged routed runoff. Only streams with a runoff rate higher than the threshold (in this case study, 0.145 m³/s) are delineated as valid streams. In other words, we neglected the intermittent streams to the upper stream reaches (Figure 5). The preprocessed stream network consists of a main stream and four tributaries (Figure 5b). The reach of each stream is defined as a polyline in a geometry file. As illustrated in Section 2.3, uniformly disaggregated groundwater discharges processed by the interface RIV2FEM were assigned to every OGS mesh node within the stream network.

### 3.5 Calibration procedure

The calibration of the coupled model follows a two-step procedure. In the first step, mHM was calibrated independent of OGS for the period from 1970 to 2005 by matching the observed runoff at the outlet of the catchment. The first 5 years were used as spin-up period to set up initial conditions in the near-surface soil zone. The calibration quality is quantified by the Nash-Sutcliffe coefficient of efficiency (NSE):

$$NSE = 1 - \frac{\sum_{i=1}^{n} |(q_m - q_s)|_i^2}{\sum_{i=1}^{n} |(q_m - \bar{q}_m)|_i^2} \tag{14}$$

where $q_s$ is the simulated discharge [L³T⁻¹], $q_m$ is the measured discharge [L³T⁻¹], and $\bar{q}_m$ is the mean of measured discharge [L³T⁻¹].

In the second step, the steady-state groundwater model in OGS was calibrated to match the long-term mean of observed groundwater levels. The long-term mean of recharge and baseflow estimated by mHM were fed to the steady-state groundwater model as Neumann boundary conditions. The calibration was performed using the software package PEST (Doherty et al., 1994). The model parameters were adjusted within a fixed interval until the value of objective function, which is the sum of weighted squared residuals of modeled and observed groundwater heads, was minimized. Specifically, the intervals of adjustable parameters were taken from the literature (Wechsung, 2005; Seidel, 2004), and the weights assigned to each observation were set uniformly to 1. The calibration result is assessed by the root-mean-square error (RMSE).

### 3.6 Model evaluation and sensitivity analysis

We used the time series of groundwater levels in 19 monitoring wells to evaluate the predictive capability of the transient model. In the transient model, hydraulic conductivities are obtained from the calibrated steady-state model. Meanwhile, the initial condition of the groundwater head is directly taken from the result of steady-state model. The Pearson correlation coefficient $R_{cor}$ and the inter-quartile range error QRE are used as two summary statistics to evaluate the predictive capability. The (relative) inter-quartile range error QRE is defined by:

$$QRE = \frac{IQ_{7525}^{md} - IQ_{7525}^{dt}}{IQ_{7525}^{dt}} \tag{15}$$

where $IQ_{7525}^{md}$ and $IQ_{7525}^{dt}$ are the inter-quartile ranges of simulations and observations, respectively.

We sought to quantify the sensitivity of groundwater flows to the different spatial pattern of recharge. For this purpose, a uniform recharge scenario was established as the reference scenario. The sensitivity analysis follows a two-step workflow. First, we calibrated the steady-state groundwater models for the two recharge scenarios independently. Second, we conducted transient simulations by assigning the same values of storage parameters, and then observed their corresponding performances in two recharge scenarios. With the exception of recharge scenario and hydraulic-conductivity values, all model parameters (e.g., specific yield and specific storage) and inputs are set to be identical in both scenarios. The mean absolute error (MAE), and the inter-quartile range error QRE are used as two summary skill scores to assess model performances in the two recharge scenarios.

## 4 Results

### 4.1 Calibration

As the first part of calibration, mHM is calibrated against discharge. The calibration results demonstrate the predictive capability of mHM in reproducing the time series of catchment discharge (Figure 6). The Nash–Sutcliffe model efficiency coefficient (NSE) is 0.88. Other fluxes, such as evapotranspiration measured at eddy-covariance stations inside this area, also shows quite reasonable correspondence to the modeled estimate (Heße et al., 2017).

In the second step, the steady-state groundwater model is calibrated against the long-term mean of groundwater heads. Table 1 shows the calibrated hydraulic conductivities in each of the geological units. The objective function of calibration, which is the sum of squared weighted residuals, converged from an initial value of 8625 m$^2$ to 464.74 m$^2$ after a total of 114 model runs. Broadly speaking, the steady-state model can plausibly reproduce the finite numbers of observed groundwater heads in the catchment. Figure 7 shows the one-to-one plot of simulated and observed groundwater heads (locations of those wells are shown in Figure 3). In general, the model is capable of reproducing spatially-distributed groundwater heads over a wide range, with an overall RMSE of 6.45 m. Most of the discrepancies between individual observations and simulations are within a reasonable range (i.e., less than 6 m). Nevertheless, some monitoring wells show larger discrepancies between observations

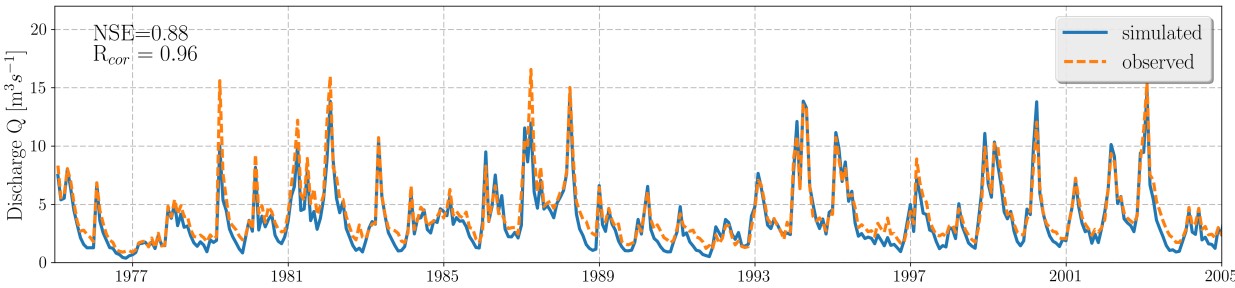

**Figure 6.** Observed and simulated monthly discharge at the outlet of the Nägelstedt catchment.

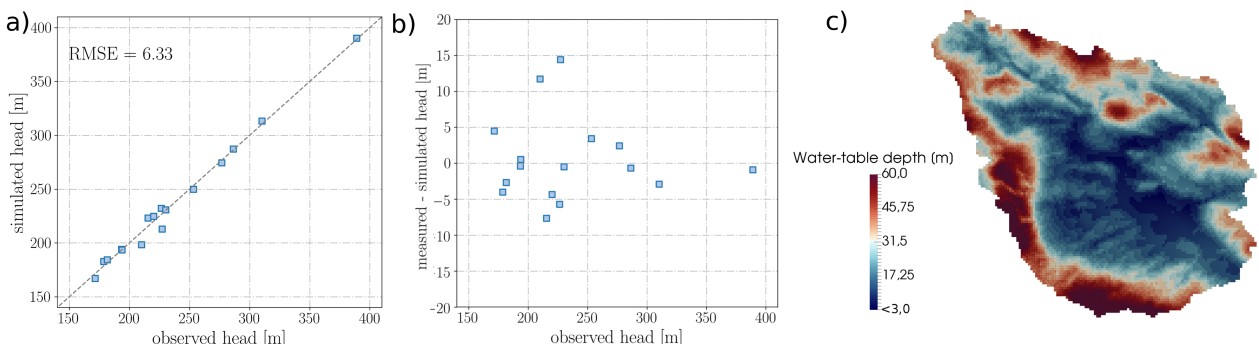

**Figure 7.** Illustration of steady-state groundwater model calibration and simulated heads. (a) Observed and simulated groundwater head (including RMSE); (b) Difference between simulated and observed head related to the observed head values; c) Simulated long term mean water table depth across the Nägelstedt catchment.

and simulations (i.e., greater than 6 m), which is due to the unknown local or even subgrid-scale properties. For the sake of simplicity, no further attempt was made to add more model complexity to improve the model fit.

The simulated depth to groundwater over the whole catchment using the calibrated hydraulic-conductivity values is shown in Figure 7c. Broadly speaking, the calibrated model reasonably reproduces the spatial groundwater table distribution. Ground-
5  water depth varies between greater than 40 m in the higher southwestern and northern mountainous areas, to less than 5 m in the central lowlands. The plausibility of steady-state simulation results can be assessed through regionalized observations of groundwater heads (Wechsung, 2005).

## 4.2  Spatio-temporal patterns of recharge and baseflow

Groundwater recharge has a spatially variable and dynamic behavior depending on the sporadic, irregular, and complex features
10  of precipitation, geological structure, and morphological features. The temporal and spatial variability of groundwater recharge and baseflow is estimated by mHM over a period of 30 years from 1975 to 2005.

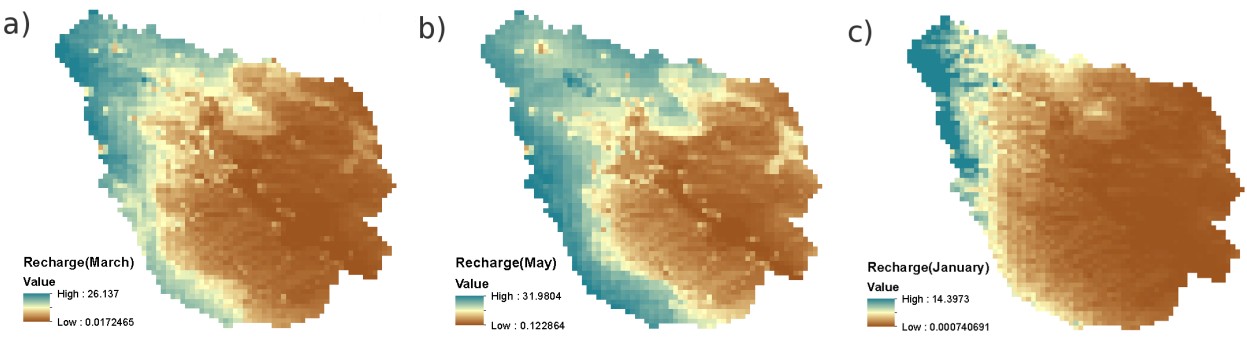

**Figure 8.** Spatial distributions of groundwater recharge in the Nägelstedt catchment (unit: mm/month) (a) in March (b) in May, and (c) in January 2005.

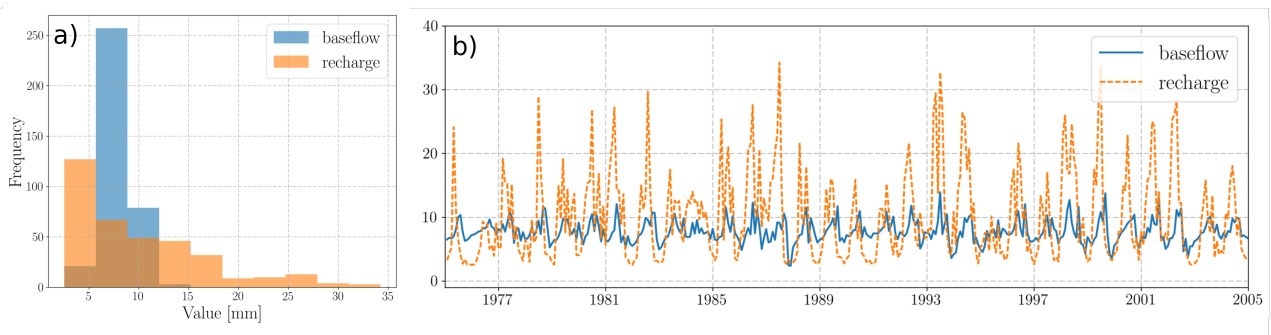

**Figure 9.** Analysis of groundwater inflow (recharge) and outflow (baseflow) over the Nägelstedt catchment. a) Distribution of groundwater balance components. b) Monthly time series of groundwater recharge and baseflow.

Figure 8 shows the spatial variability of groundwater recharge in three months: March (Figure 8a), May (Figure 8b), and January (Figure 8c). The results indicate that the location of highest recharge rate is in the upstream mountainous areas where the Muschelkalk aquifer crops out, but varies in different seasons. The maximum value of monthly groundwater recharge varies from 26 mm in March, to 51 mm in May and 14 mm January. We also evaluated the plausibility of groundwater recharge simulated by mHM through comparison to other reference datasets. At the large-scale, the groundwater recharge simulated by mHM agrees quite well with estimates from the Hydrological Atlas of Germany (Zink et al., 2017).

Figure 9a shows the distribution of monthly groundwater recharge and monthly baseflow. Over the entire year, groundwater inflow (recharge) and outflow (baseflow) are balanced, exhibiting a mean value of 8 mm/month. The difference between the two values is merely 2%. The Figure 9a, however, indicates that the distribution of monthly groundwater recharge is skewed to the right, whereas the distribution of monthly baseflow is more peaked. Figure 9b depicts the time series of groundwater recharge and baseflow, which further demonstrates that the deviation of monthly groundwater recharge is larger than the baseflow. This phenomenon further reveals the significant buffering effect of the linear groundwater storage in mHM.

**Table 1.** Main hydraulic properties used in the case study under the default mHM-generated recharge scenario.

| Geological units | Hydraulic conductivity [m/s] | | | Specific yield [-] | Specific storage [$m^{-1}$] |
|---|---|---|---|---|---|
| | Lower limit | Upper limit | Calibrated value [m/s] | | |
| km | $1.0 \times 10^{-6}$ | $5.5 \times 10^{-3}$ | $1.844 \times 10^{-5}$ | - | $1 \times 10^{-6}$ |
| ku | $1.0 \times 10^{-7}$ | $3.4 \times 10^{-4}$ | $2.848 \times 10^{-5}$ | - | $1 \times 10^{-6}$ |
| mo1 | $8.0 \times 10^{-8}$ | $2.0 \times 10^{-3}$ | $3.570 \times 10^{-5}$ | 0.10 | $1 \times 10^{-6}$ |
| mm1 | $1.0 \times 10^{-7}$ | $9.0 \times 10^{-4}$ | $3.594 \times 10^{-5}$ | - | $1 \times 10^{-6}$ |
| mu1 | $5.0 \times 10^{-9}$ | $2.0 \times 10^{-4}$ | $6.202 \times 10^{-6}$ | - | $1 \times 10^{-6}$ |
| mo2 | $1.0 \times 10^{-8}$ | $5.0 \times 10^{-4}$ | $3.570 \times 10^{-6}$ | - | $1 \times 10^{-6}$ |
| mm2 | $3.0 \times 10^{-8}$ | $9.0 \times 10^{-5}$ | $3.594 \times 10^{-6}$ | - | $1 \times 10^{-6}$ |
| mu2 | $5.0 \times 10^{-10}$ | $2.0 \times 10^{-5}$ | $6.202 \times 10^{-7}$ | - | $1 \times 10^{-6}$ |
| soil | $5.0 \times 10^{-5}$ | $1.0 \times 10^{-2}$ | $6.617 \times 10^{-5}$ | 0.10 | - |
| alluvium | $4.0 \times 10^{-5}$ | $1.0 \times 10^{-2}$ | $3.219 \times 10^{-4}$ | 0.18 | - |

### 4.3 Model evaluation against dynamic groundwater heads

In this subsection, the head observations of several monitoring wells in the catchment were used to evaluate the model performance. We analyzed discrepancies between the modeled and observed groundwater heads by subtracting long-term mean values, $\overline{h}_{mod}$ and $\overline{h}_{obs}$. Four model-skill scores including the mean value, the median value, the Pearson correlation coefficient $R_{cor}$, and the inter-quartile range error, QRE, are used to evaluate the model performance.

Six wells with different geological and morphological properties were chosen as samples to exhibit the model performance (Figure 10). Specifically, well W10 is located in the northern uplands and is near the main stream, whereas well W1 is located in the southwestern lowlands. As can be observed from Figure 10, they provide good fits between simulated and observed heads, with a $R_{cor}$ of 0.87 and 0.76, and QRE of -23.34% and -1.65%, respectively. Well W17 is located in the Lower Keuper unit, while well W16 is located in the upper Muschelkalk formation. In these two monitoring wells, the simulations are highly correlated with observations with high values of $R_{cor}$ (0.71 and 0.82), in spite of their different geological properties (Figure 10). The simulation results at monitoring well W13 (located in the northern mountainous area) and W7 (located at the northern upland) also exhibit good correspondence with the observations (Figure 10). In general, the model is capable of capturing the historical trends of groundwater dynamics, even though the mean values of simulations and observations may deviate to some extent. Due to the limited spatial resolution and complex hydrogeological structure, this degree of discrepancy is acceptable.

### 4.4 Model sensitivity to different recharge scenarios

As described in Section 3.6, a reference recharge scenario (RR), i.e., a spatially uniform recharge scenario, is set up to assess the effect of spatial patterns of recharge on groundwater heads. In this uniform recharge scenario, RR, the steady-state groundwater model, was re-calibrated using the long-term mean of spatially uniform recharge (Table 2). For the purpose of showing

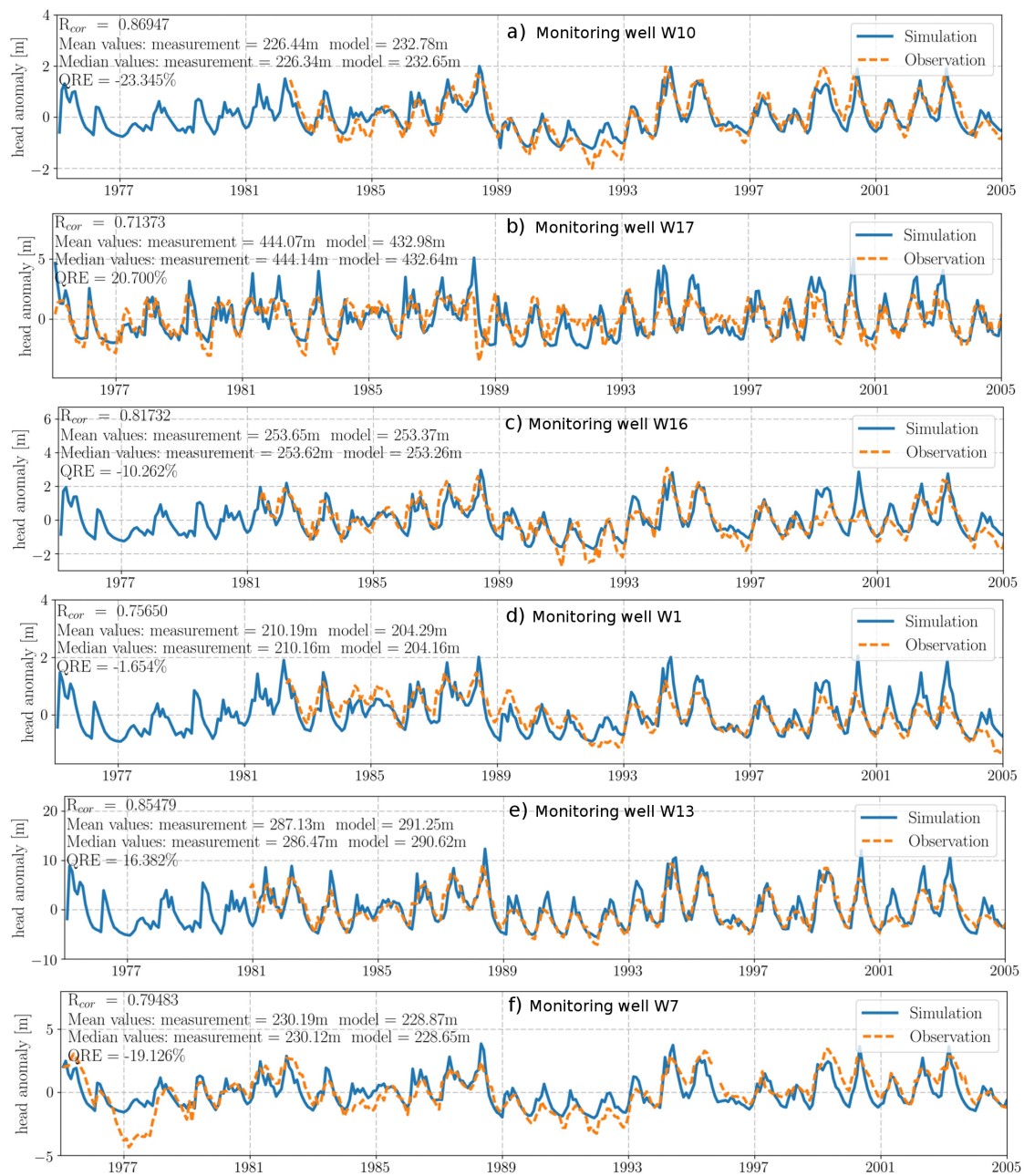

**Figure 10.** Comparison between measured (green dashed line) and simulated groundwater head anomalies (blue solid line). (a) W10 is located in uplands, near a stream. (b) W17 is located in a mountainous area. (c) W16 is located at a hillslope in the northern upland. (d) W1 is located in the lowland. (e) W13 is located in the northern mountains. (f) W7 is located in the northern upland.

**Table 2.** Hydraulic properties used in the uniform-recharge scenario (RR).

| Geological units | km | ku | mo1 | mm1 | mu1 | mo2 | mm2 | mu2 | soil | alluvium |
|---|---|---|---|---|---|---|---|---|---|---|
| Hydraulic conductivity | 5.023 | 6.216 | 8.608 | 2.990 | 5.316 | 8.604 | 2.997 | 5.317 | 5.239 | 7.302 |
| [m/s] | $\times 10^{-5}$ | $\times 10^{-5}$ | $\times 10^{-5}$ | $\times 10^{-5}$ | $\times 10^{-6}$ | $\times 10^{-6}$ | $\times 10^{-6}$ | $\times 10^{-7}$ | $\times 10^{-5}$ | $\times 10^{-4}$ |

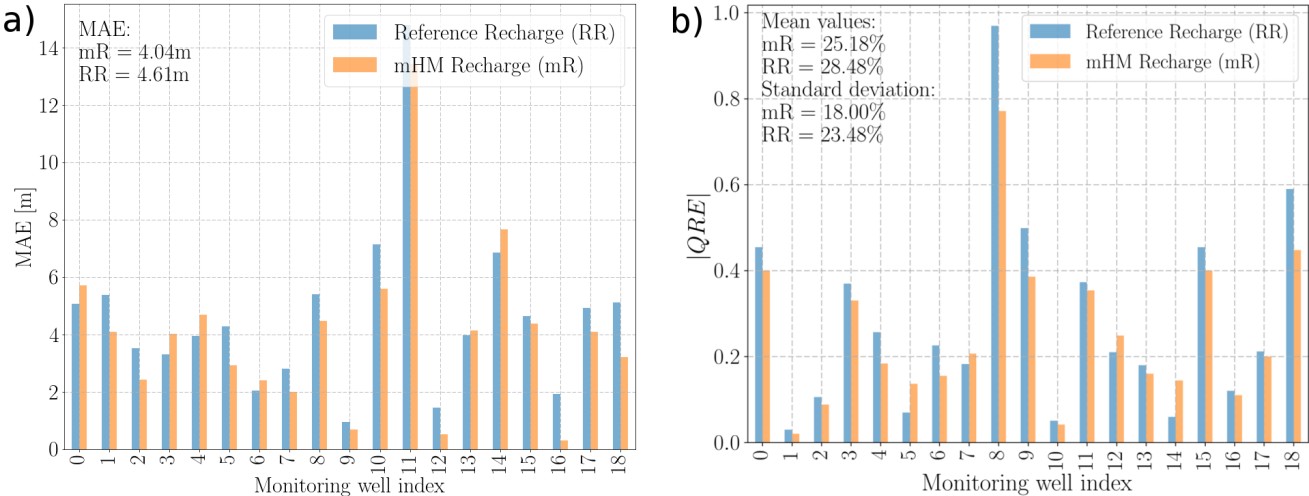

**Figure 11.** Barplots of a) the mean absolute error MAE and b) the absolute inter-quartile range error |QRE| in all monitoring wells in two recharge scenarios.

discrepancies between two recharge scenarios, we compared the values of MAE and |QRE| at each monitoring well between the spatially distributed recharge, mR, and the uniform recharge, RR (Figure 11). The mean value and the median value of |QRE| were also calculated and are shown in Figure 11. Figure 11a indicates that the MAE using the spatially distributed recharge mR (4.04 m) is lower than that using the uniform recharge RR (4.61 m). Considering that the only difference between the two recharge scenarios is their spatial patterns, we conclude that accounting for spatially-distributed recharge provides a moderate

5 improvement in the model.

Figure 11b shows the absolute values of inter-quartile range error (|QRE|) in simulations using the two recharge scenarios (mR and RR). We found that the deviation of |QRE| is significantly larger than $R_{cor}$, i.e., the |QRE| in two wells are abnormally higher than the other wells. The higher values of |QRE| at W8 and W18 may be caused by their proximity to model boundaries, as the two wells are located either near a river or near the catchment perimeter. This deviation indicates that accurate quantifi-

10 cation of the amplitude of head fluctuations at certain locations is difficult, which may be due to the proximity of boundaries or complex local topography and geology. Nevertheless, 16 out of 19 wells exhibit low inter-quartile range errors, with the values of |QRE| in a range of ±40% in the spatially distributed mR scenario. We also observe a smaller mean and standard deviation of |QRE| in the spatially distributed mR than in the uniform scenario, RR. The 19 chosen monitoring wells cover the geologi-

15 cal units of the alluvium, Keuper, and Muschelkalk, and range from high mountains to lowlands across the catchment. These

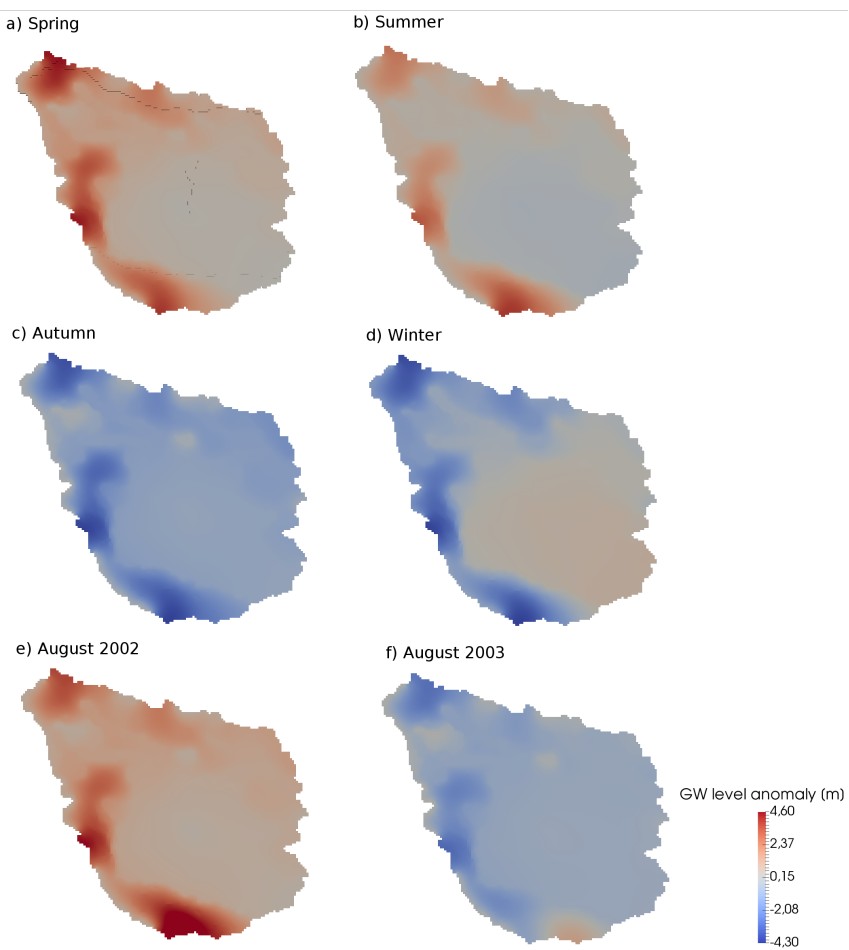

**Figure 12.** Seasonal variation of spatially-distributed groundwater heads by their anomalies after removing the long-term mean groundwater heads (unit: m). a) Long-term mean groundwater head distribution in spring; b) Long-term mean groundwater head distribution in summer; c) Long-term mean groundwater head distribution in autumn; d) Long-term mean groundwater head distribution in winter; e) Monthly mean groundwater head distribution in the wet season (August 2002); f) Monthly mean groundwater head distribution in the dry season (August 2003).

results demonstrate the promising modeling capability of the model and highlight the moderately better historical matching when using a spatially distributed pattern of groundwater recharge.

Figure 12 illustrates the seasonality of groundwater heads by showing the spatial distribution of groundwater heads averaged over the spring, summer, autumn, and winter seasons, respectively. A strong spatial variability can be observed. For example, the fluctuation amplitudes of groundwater heads in the northern, eastern, and southeastern mountainous areas are larger than in the central plains area. In order to illustrate predicted groundwater levels and droughts caused by extreme climate events, we selected a meteorologically wet month (August 2002) and a meteorologically dry month (August 2003), and show the

corresponding variations of groundwater heads in Figures 12e and 12f. In general, the groundwater heads in the wet season are higher than the long term mean values (Figure 12e). The variation of groundwater heads in the dry season, however, shows a strong spatial variability. Such a strong spatial variability of groundwater heads variation has also been reported by Kumar et al. (2016).

## 5   Discussion and conclusions

Our simulation results demonstrate that the coupled model mHM#OGS v1.0 can generally reproduce groundwater-head dynamics very well. It is also able to reasonably reproduce fluctuation amplitudes of groundwater heads, although with less accuracy. The simulation results also reveal that the stochastical and physically-based representations of groundwater dynamics can be intrinsically linked on the condition that the geometry and geological structure of groundwater aquifer are reasonably characterized. Compared to the good predictive capability of capturing the general trend behavior, the amplitude of head time series is hard to reproduce. This might be because local geological formations in the vicinity of monitoring wells may significantly alter local groundwater flow behavior, and thus further affect groundwater head fluctuations.

The results of this study demonstrate the successful application of the well-established hydrologic model, mHM, in estimating spatially heterogeneous groundwater recharge and baseflow at a regional scale. At a spatial scale of $10^3$ km$^2$ (the scale in this study), the distributed recharge estimated by mHM is superior to using homogeneous recharge. mHM has been successfully applied at the continental scale covering entire Europe (Thober et al., 2015; Kumar et al., 2013b; Rakovec et al., 2016b; Zink et al., 2017). The successful application of the coupled model in this study suggests a huge potential for extending the applicability of mHM#OGS v1.0 to a larger-scale (e.g., $10^4$ - $10^6$ km$^2$) or even a global scale.

The results of this study demonstrate a viable strategy for improving classic meso- to large-scale distributed hydrologic models, such as the current version of mHM (Samaniego et al., 2010; Kumar et al., 2013b), VIC (Liang et al., 1994), PCR-GLOBWB (Van Beek and Bierkens, 2009), WASMOD-M (Widén-Nilsson et al., 2007). These distributed hydrologic models do not calculate spatio-temporal groundwater heads and are therefore unable to represent groundwater head dynamics in their groundwater compartment. The physical representation of groundwater flow is, however, relevant in future regional-scale and possibly global hydrologic models to accurately determine travel times, solute export from catchments, and water quality in rivers (Botter et al., 2010; Benettin et al., 2015; Van Meter et al., 2017). The coupled model mHM#OGS v1.0 also offers the potential for predicting groundwater drought in analyzing the dynamic behavior of groundwater heads. Thus, it could be a useful tool for understanding groundwater anomalies under extreme climate conditions (Kumar et al., 2016; Marx et al., 2017).

For example, building on previous work of Heße et al. (2017), who calculated Travel Time Distributions (TTDs) using mHM, we can now expand the range of their work to the complete critical zone, which is important for comprehensively understanding particle (e.g., pollutant) transport behavior and the historical legacy in soil zone and groundwater storage (Basu et al., 2010; Beniston et al., 2014). mHM#OGS v1.0 fits well with the long-term simulation of nitrogen transport in the terrestrial water cycle. The coupled model is also able to evaluate surface water and groundwater storage changes under different meteorological forcing conditions, which allows the comprehensive evaluation of hydrologic response to climate changes (e.g., global

warming). Additionally, OGS demonstrates its capability in addressing thermo-hydro-mechanical-chemical (THMC) coupling processes in large-scale hydrologic cycles (not reflected in this study), which is significant for a wide range of real-world applications, including nutrient circulation, salt water intrusion, drought, and heavy metal transport (Kalbacher et al., 2012; Selle et al., 2013; Walther et al., 2014, 2017).

In addition to improving the predictive capabilities of mHM, we can also demonstrate some improvements for the groundwater model OGS. Our results showed a modest improvement using mHM generated recharge compared to a simpler, uniform recharge rate. We currently gain a strong advantage for the description of the top boundary condition, i.e. the recharge, which is temporal and spatially variable through the input of mHM. Even more, the recharge fluxes provided are based on mHM's phenomenological process description, which significantly better describes the surface level recharge fluxes than common

approaches through recharge rates derived by empirical relations.

    In this study, we have focused our efforts on extending the applicability of mHM from surface hydrology to subsurface hydrology by a simple one-way coupling. Consequently, we do not account for any feedback between river and groundwater head fluctuations. This approach is parsimonious and numerically efficient, meanwhile fully preserves the well-tested parameterization algorithm in mHM. Unlike two-way coupling, the one-way coupling described here allows the user to expand the

abilities of mHM without sacrificing any of its well-known and well-established properties. Nevertheless, in a next step, we will devote to incorporate a full, two-way coupling using the next version of the mHM#OGS model. The main limitation of one-way coupling is that the effects of a shallow depth to groundwater on actual ET, maintained by lateral groundwater flow, cannot be explicitly addressed. However, the dynamic interactions between overland flow and groundwater flow, as well as between soil moisture dynamics and groundwater dynamics can explicitly be modeled and investigated using a full coupling scheme. This

approach is open to a broader spectrum of calibration options, such as calibration by remotely sensed soil moisture data.

    In conclusion, we can state that the coupled model mHM#OGS v1.0 retains the predictive capability of mHM for discharge volumes. In addition, it is capable of reproducing groundwater head dynamics. The simulation results indicate a promising predictive ability, confirmed by calibration and comparison to observed discharge and groundwater heads. Based on the historical match of discharge and groundwater heads in the case study, we conclude that the coupled model mHM#OGS v1.0 is

a valuable tool for addressing many challenging problems in the field of water management, including pollutant transport and legacy, climate change, and groundwater drought.

*Code and data availability.*   The mesoscale Hydrologic Model mHM (current release: 5.7) is an open-source community software and can be accessed from several mirrored repositories: SVN: http://www.ufz.de/index.php?en=40114; GitLab: https://git.ufz.de/mhm; GitHub: https://github.com/mhm-ufz. The modified source code of OGS5 can be freely acquired via the following link: https://github.com/UFZ-MJ/OGS_

mHM.git. The model interface GIS2FEM and RIV2FEM can be freely acquired via the following link: https://github.com/UFZ-MJ/OGS_mHM/tree/master/UTL/GIS2FEM and https://github.com/UFZ-MJ/OGS_mHM/tree/master/RIV2FEM.

    The input files of the case study in Nägelstedt catchment can be found in the Github repository: https://github.com/UFZ-MJ/OGS_mHM/tree/master/test_case. The dataset used in the case study can be found in the Github repository: https://github.com/UFZ-MJ/OGS_mHM/tree/master/data.

*Acknowledgements.* This research received funding from the Deutsche Forschungsgemeinschaft via Sonderforschungs- bereich CRC 1076 AquaDiva. We kindly thank Sabine Sattler from Thuringian State office for the Environment and Geology (TLUG) for providing the geological data. We kindly thank our data providers: the German Weather Service (DWD), the Joint Research Center of the European Commission, the Federal Institute for Geosciences and Natural Resources (BGR), the European Environmental Agency (EEA), the European Water Archive (EWA), and the Global Runoff Data Centre (GRDC). We thank the two reviewers Prof. Olaf Cirpka and Dr. Edwin Sutanudjaja very much for their comprehensive reviews.

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
