# Peer review of "Improved regional scale groundwater representation by the coupling of the mesoscale Hydrologic Model (mHM v5.7) to the groundwater model OpenGeoSys (OGS)"

_Geoscientific Model Development, 2017_

## Author Comment (AC1) · 26 Sep 2017

There is a spelling error in the title: "OpeneGeoSys" should be corrected as "Open-GeoSys".

---

## Short Comment (SC1) · 29 Sep 2017

I have read the paper with great interest and find this kind of developments within SW-GW coupling very relevant and necessary. I generally find the paper very interesting and a good step in the right direction. I especially encourage the efforts related to variable time stepping and grid resolutions between compartments. I would however encourage the authors to be more specific about the coupling and limitations hereof. As I read it there is no feedback from OGS to mHM, meaning that mHM is merely used to calculate a distributed boundary condition of groundwater recharge to OGS, which could have been done using separate models? Or am I wrong? There is currently

no coupling between groundwater and soil moisture/evapotranspiration or baseflow? I find the discussion part interesting when it comes to actual full coupling. This would make the potential for model application far greater since it would enable simulations of the impact of horizontal GW flow on surface water and the impact of groundwater levels on GW-SW interactions. This again would make the model useful for evaluating the effect of GW pumping on surface water flow. One could ask, what is the purpose of a regional scale ground water model from a water resources perspective if it does not include the interaction with surface water? I suggest that the authors: • Make it very clear from the beginning of the manuscript which kind of "coupling" is performed • Provide more details on the OGS code, is it a fully integrated 3D variably saturated code or a pure saturated GW code? • That the discussion section about full coupling is expanded by reflecting more on what that would require (e.g. regarding flexible time stepping and grid resolution etc.) and what sort of application that could benefit from such a development. Also referencing other fully coupled codes. • Discuss what kind of water resources issues this kind of regional scale model (mHM#OGS) could help solve.

Thanks for an interesting contribution, that I hope to see in GMD and hopefully also a follow-up paper including a full GW-SW coupling which would really excel the potential of such a model system.

Minor comment: Figure 8 need to include specifications of a) and b) and the figure caption needs to explain what the blue and red plots represent. Also I think you should avoid adding the Rcor values for groundwater heads, since they are meaningless in a topographically varying catchment. Stick to the RMSE.

---

## Short Comment (SC2) · 5 Oct 2017

Dear authors,

in my role as Executive editor of GMD, I would like to bring to your attention our Editorial version 1.1:

http://www.geosci-model-dev.net/8/3487/2015/gmd-8-3487-2015.html

This highlights some requirements of papers published in GMD, which is also available on the GMD website in the 'Manuscript Types' section:

http://www.geoscientific-model-development.net/submission/manuscript_types.html

In particular, please note that for your paper, the following requirement has not been met in the Discussions paper:

- "The main paper must give the model name and version number (or other unique identifier) in the title."

Please add the version numbers of mHM and OGS in the title upon your revised submission to GMD.

Yours,

Astrid Kerkweg

---

## Referee Comment (RC1) · Anonymous Referee #1 · 27 Oct 2017

The authors present a coupling approach for a land surface hydrologic and groundwater flow model, mHm and OGS respectively. The manuscript contains sections on the coupling, model setup over a real catchment and verification of the results. The model coupling is not explained appropriately and it's not clear, whether the coupling approach satisfies the current state-of-the-art published in GMD. Based on the provided explanation, the results can not be assessed unfortunately.

Introduction The introduction is incomplete and misses some of the most important and heavily cited references of integrated models and modeling studies of the terrestrial water cycle. Apparently the authors are not aware of the state-of-the-art. Proper citation

of the mentioned models is missing. Is the sole goal of the introduction to promote the work of the co-authors (e.g. statement p 3, l 12-15 and citations throughout)?

Model description Section 2.1 and 2.2 must be expanded. At least, the reader must get some idea about the basic principles that are used to model the different processes mentioned in passing, in order to assess the validity of the coupling. In section 2.3, figure 1b, suggests one-way coupling only i.e. mHm provides "groundwater recharge and base flow as boundary conditions to mHm" (p 3, l 16-17). Since mHm does not include groundwater, how can the calculation of these fluxes be mechanistic (p 3, l 15), because groundwater recharge strongly depends on the dynamics of the water table? Thus, the scarce information provided in this section in combination with the statements in the introduction are misleading to the reader.

Section 2.3.2 with the title "Boundary condition-based coupling" provides the basic equations, yet leaves the reader wondering how the coupling is really done. Something is said about the exchange of fluxes via qe and qe' (p 7, l 3), but these are sources not boundary fluxes. What is equation 2? The upper boundary condition for the groundwater flow model? Shouldn't the coupling be performed via equation 2 as promised in the section title? In addition, the authors state that "the coupling interface converts time series of variables and fluxes to Neumann boundary conditions...". How does that fit in? This reader is left confused.

Figure 2 is not instructive. What is GIS2FEM doing? Interpolating? How does the coupling work in the vertical direction for each column? As I understand, mHm has a fixed column depth. Can the water table rise into the column along e.g. river corridors? And where does the baseflow go in OGS? How is groundwater storage in mHm (p 7, l 9-10) related to OGS? There is apparently no backward exchange with mHm due to baseflow and exchange with river networks, and no capillary rise. This reader is left confused.

On p 7, l 17-18, what do the authors mean by conversion between volumetric flux,

specific flux and water head? Where in the coupling is this conversion required and why does the cell sizes need to be adjusted (there is actual re-gridding going on)?

From table 2 it appears that in the author's eyes, coupling and integrated modeling of the terrestrial water cycle simply means to pass groundwater recharge values from a 1D hydrologic land surface scheme to a steady state groundwater flow model and return a head value back as some lower (boundary) condition for the hydrologic scheme (not indicated in figure 1). I feel, in the geosciences, we moved beyond this type of approach quite some time ago.

The description of the study area and model setup, calibration etc. belong into a separate section.

The results can not be assessed unfortunately, because of the poor explanation of the applied modeling and coupling techniques.

Language and grammar require considerable improvement.

––––––––––––––––––––––––

---

## Author Comment (AC2) · 2 Nov 2017

On behalf of all authors, I sincerely thank Dr. Stisen for his comments. I really appreciate his effort in reading and understanding our work. We have prepared a response to each of the reviewer's comments, and have suggested how we will incorporate these suggestions into a revised manuscript.

With regards to the main comments:

1. I have read the paper with great interest and find this kind of developments within SW/GW coupling very relevant and necessary. I generally find the paper very interesting and a good step in the right direction. I especially encourage the efforts related to variable time stepping and grid resolutions between compartments. I would however encourage the authors to be more specific about the coupling and limitations hereof.

This paper focuses on two things. The first one is that the authors build a practical workflow in which mHM and OGS are dynamically coupled. This paper presents the first attempt in coupling two codes with distinct usages (mHM for predicting mesoscale catchment runoff, OGS for solving multi-physical problems in porous media), structures (mHM written in Fortran 2008 and OGS written in C++) and even philosophies (mHM seeks for a good prediction ability across multi-scale catchments in a computationally efficient way, OGS solves computationally-expensive non-linear PDEs using Finite Element Method). The second main point is we demonstrate that the groundwater head dynamics can be well characterized in the mesoscale catchment by the coupled model. The effectiveness of spatially distributed heterogeneous groundwater recharge is also tested hereby.

2. As I read it there is no feedback from OGS to mHM, meaning that mHM is merely used to calculate a distributed boundary condition of groundwater recharge to OGS, whichcould have been done using separate models? Or am I wrong? There is currently no coupling between groundwater and soil moisture/evapotranspiration or baseflow? I find the discussion part interesting when it comes to actual full coupling. This would make the potential for model application far greater since it would enable simulations of the impact of horizontal GW flow on surface water and the impact of groundwater levels on GW-SW interactions. This again would make the model useful for evaluating the effect of GW pumping on surface water flow.

In the current version of mHM#OGS, the mHM first calculates the daily spatially-distributed recharge and baseflow, then feed the two time series of fluxes to OGS through an interface which converts unit, adjusts grid and time step sizes automatically. Therefore mHM does not only provide groundwater recharge, but also provide baseflow as a boundary condition of groundwater model (please see the Figure 7 in the manuscript). Through calibration against groundwater heads (manually or automatically), one can get a plausible estimation of groundwater storage, which is very important and always missing in typical bucket-type hydrological models. The reasonable quantification of K values and storage is the basis of several important scientific questions such as storage-runoff correlation, groundwater drought, and contaminant legacy.

3. One could ask, what is the purposeof a regional scale ground water model from a water resources perspective if it does not include the interaction with surface water? I suggest that the authors: ă˘A´c

Make it very clear from the beginning of the manuscript which kind of "coupling" is performed.

I agree with the Dr. Stisen that there are some limitations of the current offline coupling approach as it does not include the feedback of groundwater dynamics to surface processes. I would also like to draw attention to the starting point of our initiative in coupling those two codes. The starting point is on the hydrogeological side, rather than on the hydrological side. The current coupling model, as the first attempt, is designed to reproduce the dynamics of groundwater head and velocity and calibrate K values and groundwater storage using the boundary condition given by mHM. The feedback to surface water processes is another topic, and is the follow-up process only if the K values, groundwater storage are carefully calibrated and groundwater head dynamics are well reproduced.

Moreover, the current coupling method can help to answer a couple of important scientific questions. For example, Kumar et al [1] have demonstrated that the Standardized Precipitation Index (SPI) has a limited applicability and low reliability in characterizing groundwater drought. Our model can be a useful tool in predicting groundwater drought & flood under different climate conditions (please check Figure 11 and 13 in the manuscript). Moreover, the coupled model can be used to quantify the catchment scale legacy nitrogen stores in groundwater reservoirs. Recent research shows that a large portion of legacy nitrogen can be older than 10 years [2]. The current version of mHM#OGS fits well with the long-term simulation of nitrogen transport in beneath-atmosphere water cycle owing to its nested time stepping.

4. Provide more details on the OGS code, is it a fully integrated 3D variably saturated code or a pure saturated GW code?

I also agree with the Dr. Stisen that I should introduce more about OGS in terms of its capability in simulating Richards flow. OGS is 3D variably saturated code. Its capability in simulating variably saturated zone flow has been verified [3,4]. OGS is involved in a model inter-comparison project and is tested based on a series of benchmark problems [5]. Due to the fact that the overall aim of the current model is to reproduce groundwater head, the unsaturated zone flow is less-important and could be simulated in a conceptualized way using mHM. Nevertheless, I fully agree with the author that the variably saturated flow should be added into the next version of the coupled model.

5. Minor comment: Figure 8 need to include specifications of a) and b) and the figure caption needs to explain what the blue and red plots represent. Also I think you should avoid adding the Rcor values for groundwater heads, since they are meaningless in a topographically varying catchment. Stick to the RMSE.

I thank the Dr. Stisen for his minor comment which are very helpful. The Figure 8 a) and b) are actually showing different goodness of matching under two different recharge scenarios. I will stick to RMSE in the revised paper following your suggestion.

**References**

[1] Kumar, R., Musuuza, J. L., Van Loon, A. F., Teuling, A. J., Barthel, R., Ten Broek, J., Mai, J., Samaniego, L. and Attinger, S.: Multiscale evaluation of the Standardized Precipitation Index as a groundwater drought indicator, Hydrol. Earth Syst. Sci., 20(3), 1117–1131, doi:10.5194/hess-20-1117-2016, 2016.

[2] Van Meter, K. J., Basu, N. B. and Van Cappellen, P.: Two centuries of nitrogen dynamics: Legacy sources and sinks in the Mississippi and Susquehanna River Basins, Global Biogeochem. Cycles, 31(1), 2–23, doi:10.1002/2016GB005498, 2017.

[3] Wang W, Rutqvist J, Görke UJ, Birkholzer JT, Kolditz O. Non-isothermal flow in low permeable porous media: a comparison of Richards' and two-phase flow approaches. Environmental Earth Sciences. 2011 Mar 1;62(6):1197-207.

[4] Kolditz O, Shao H, Wang W, Bauer S. Thermo-hydro-mechanical-chemical processes in fractured porous media: modelling and benchmarking. Springer International Pu; 2016.

[5] Maxwell, R. M., Putti, M., Meyerhoff, S., Delfs, J.-O., Ferguson, I. M., Ivanov, V., Kim, J., Kolditz, O., Kollet, S. J., Kumar, M., Lopez, S., Niu, J., Paniconi, C., Park, Y.-J., Phanikumar, M. S., Shen, C., Sudicky, E. A. and Sulis, M.: Surface-subsurface model intercomparison: A first set of benchmark results to diagnose integrated hydrology and feedbacks, Water Resour. Res., 50(2), 1531–1549 [online] Available from: http://onlinelibrary.wiley.com/doi/10.1002/2013WR013725/full (Accessed 18 August 2016), 2014.

---

## Author Comment (AC3) · 11 Nov 2017

We thank the referee reviewer very much for reading our work and insightful comments. Those comments really let us know the unclear part of our manuscript and help us a lot to improve our manuscript.

Since the manuscript is still under discussion phase, we would like to give the quickest feedback to explain some most important issues which are missing in the manuscript. For the details of the response comment, please check the supplement.

Please also note the supplement to this comment:

[Figure]

https://www.geosci-model-dev-discuss.net/gmd-2017-231/gmd-2017-231-AC3-supplement.pdf

**Supplement:**

**Responses to Anonymous Referee #1**

We thank the referee reviewer very much for reading our work and insightful comments. Those comments really let us know the unclear part of our manuscript and help us a lot to improve our manuscript.

Since the manuscript is still under discussion phase, we would like to write this quick response to explain some most important issues which are missing in the manuscript. As comments from the other referee reviewers are not ready yet, we are not going to present the fully revised manuscript in this response. Of course, once the other reviewers' comments are available, we will incorporate those modifications into the final revision.

Before we reply to any specific questions of the comments, we would like to clarify two points.

The first point is that we would like to explain the linear groundwater reservoir in mHM, which was not directly explained in the manuscript since it has been included in the references(Kumar et al., 2013; Samaniego et al., 2010). mHM contains a linear reservoir to generate daily baseflow (please see Figure 1). The generated baseflow of each grid are further routed into streams using Muskingum-Cunge method. In the coupled model mHM#OGS, we take spatially distributed recharge and routed baseflow generated by the linear reservoir, then feed these two boundary sources to GIS2FEM (the coupling interface to convert unit and adjust time step), and then to OGS as upper boundary conditions. The baseflow is still calculated by the linear reservoir in mHM and routed into runoff (please see Figure 1). We have now noticed that the detailed explanation of the linear groundwater reservoir is essential and will include it into the revised manuscript.

The second point is that we are not aiming to develop a fully physically-based model. We are not aiming to study the mechanistic interaction of soil-zone processes and the groundwater heads. Instead, we are aiming to develop an open-source regional-scale model which can predict catchment runoff and groundwater head dynamics simultaneously, while preserves all existing and well-tested mHM features, e.g., the parameterization scheme (Kumar et al., 2013; Rakovec et al., 2016; Samaniego et al., 2010, 2017).

[Figure]

Figure 1 mHM#OGS as an approach into realization of groundwater head

Major comments

1. The authors present a coupling approach for a land surface hydrologic and ground water flow model, mHM and OGS respectively. The manuscript contains sections on the coupling, model setup over a real catchment and verification of the results. The model coupling is not explained appropriately and it's not clear, whether the coupling approach satisfies the current state-of-the-art published in GMD. Based on the provided explanation, the results cannot be assessed unfortunately.

Thank you for your comment. Enabling the reader to independently reproduce the results is an important aspect of the publishing process of GMD. To improve that part, we will significantly revise the model section in order to make our approach more clear to the reader and avoid misconception on our work. We will also provide a fully accessible code, a test example together with all needed data in the Github repository.

2. Introduction The introduction is incomplete and misses some of the most important and heavily cited references of integrated models and modeling studies of the terrestrial water cycle. Apparently the authors are not aware of the state-of-the-art. Proper citation of the mentioned models is missing. Is the sole goal of the introduction to promote the work of the co-authors (e.g. statement p 3, l 12-15 and citations throughout)?

Thank you for your insights. We will revise the whole introduction section accordingly and cite all the up-to-date papers properly.

To better convey these points and avoid possible future misunderstanding, we will revise the introduction section in manuscript accordingly. In addition, we further expand our literature review by properly referencing integrated surface/subsurface hydrologic models (ISSHMs) such as InHM (Smerdon et al., 2007; VanderKwaak and Loague, 2001), Parflow (Maxwell et al., 2015), tRIBS ((Ivanov et al., 2004), CATHY (Camporese et al., 2010), GSFLOW (Hunt et al., 2013; Markstrom et al., 2008), HydroGeoSphere (Hwang et al., 2014; Therrien et al., 2010), MIKE SHE (Graham and Butts, 2005), MODHMS (Panday and Huyakorn, 2004; Phi et al., 2013), GEOtop (Rigon et al., 2006), IRENE (Spanoudaki et al., 2009), CAST3M (Weill et al., 2009), PIHM (Kumar et al., 2009; Qu and Duffy, 2007) and PAWS(Shen and Phanikumar, 2010), in the revised manuscript. The coupled land surface / groundwater models (CLSGMs) include ParFlow-CLM (Ferguson and Maxwell, 2010; Maxwell and Kollet, 2008; Maxwell and Miller, 2005; Rihani et al., 2010), tRIBS + VEGGIE (Ivanov et al., 2008, 2010), SWAT and MODFLOW (Guzman et al., 2015; Kim et al., 2008), PCR-GLOBWB-MOD (Sutanudjaja et al., 2014), SWMM-OGS (Delfs et al., 2012). Nevertheless, we will revise this section in order to convey comprehensive information of the state-of-the-art science.

Next, we would like to clarify that the within the context of our manuscript a "coupled model" is not the same as a "physically-based" or "mechanistic" integrated model. We will include this point into the storyline in the revised manuscript. What we want to develop is a hybrid model that is using two different modeling paradigms which can be easily applied in regional and continental scale, rather than a mechanistic integrated model. Our reasons for this decision is that more conceptual process-based models like mHM or Noah-MP are good at predicting quantities like discharge but are highly conceptualized and there suffering from interpretability of certain processes (e.g., base flow and interflow components). More mechanistic models like Parflow and HydroGeoSphere are highly interpretable but show consistently worse performance when predicting runoff (Paniconi and Putti, 2015). To the best of our knowledge, the skill of simulating groundwater head dynamics at regional scale of mechanistic models are always neglected and seldom assessed by the data (e.g. GW head, tracer). At the larger scale, the assessment of modeled groundwater heads dynamics can only be found in very few publications (De Graaf et al., 2015; Sutanudjaja et al., 2011).

[Figure]

Figure 2 Different questions and challenges in surface and subsurface hydrology

The above mentioned different abilities of more phenomenological models (e.g., mHM, Noah-MP, etc) vs. the more mechanistic models (e.g., Parflow, Hydrogeosphere, etc.) is caused by the different challenges that are posed by the different compartments of the terrestrial water cycle. One of the main challenges in the surface & near-surface storage is process uncertainty, with the fact that processes like ET, land use, land cover, snow pack, etc. are extremely complex. The process uncertainty decreases as it goes deeper and deeper into the subsurface storage. In subsurface storage, hydrological processes are under Darcy's law and therefore conceptually simpler. Meanwhile, the data uncertainty becomes more significant in deep subsurface storage than in shallow storage (see schematic in Figure 2). Therefore, proper conceptualization is needed in the shallow storage in order to deal with this process uncertainty (please see schematic in Figure 2). Owing to this point, mHM was developed as a bucket-type model to better deal with this process uncertainty by optimally leveraging the information content in the discharge data. On the other hand, OGS is a mechanistic model, i.e., it has a very low process uncertainty but large amount of data uncertainty. It is therefore optimally suited to model processes in the deeper subsurface. To use the strengths and weaknesses of both these modeling concepts, we decided to separate our modeling domain into these two compartments, a strategy that is very common in hydrology (Benettin et al., 2015; Bertuzzo et al., 2013; Botter et al., 2010; van der Velde et al., 2015), and use this different modeling paradigms for each compartment.

We will also add the following two paragraphs into the revised manuscript:

At the larger, i.e., regional scale, most of the mechanistic integrated models are based on a continuity of pressure and flux on the SW/GW interface, while the momentum balance condition is always missing (Paniconi and Putti, 2015). The runoff is generally normalized as "storage-dependent runoff" by solving Richards equation, and the grid-wise generated runoffs are routed by a routing algorithm. These models

can principally simulate the dynamic interaction of different processes with SW/GW components, e.g., the interaction of soil moisture and GW head (Cuthbert et al., 2013; Maxwell and Miller, 2005; Rihani et al., 2010; Sutanudjaja et al., 2014), the storage-runoff correlation (Fang and Shen, 2017; Huntington and Niswonger, 2012; Koirala et al., 2014; Liang et al., 2003; VanderKwaak and Loague, 2001), and the dynamical interaction between ET and GW head (Chen and Hu, 2004; Koirala et al., 2014; Yeh and Eltahir, 2005).

In constract to that, in this study, we present a one-way coupling model mHM#OGS and focuses on the representation of Infiltration-Excess Recharge (IER) and Linear Baseflow (LB) through a case study of a mesoscale catchment. The basic scientific question we want to answer is: Can spatially distributed groundwater heads and their dynamics be reasonably captured by expanding on the abilities of a phenomenological model like mHM at the regional scale? Based on the case study, we would conclude that this expansion was successful since in addition to predicting discharge, our coupled model is also able to predict head measurements as well. Since our focus is the predictive accuracy of mHM (compared to interpretability and inference), we consider the physical plausibility of the coupling of recharge and baseflow to be a means to that end and not an end in itself. Improving the plausibility of these processes will, if done right, also lead to higher predictive power. We will elaborate on these points on more details in our answers to Comment 7.

3. Model description Section 2.1 and 2.2 must be expanded. At least, the reader must get some idea about the basic principles that are used to model the different processes mentioned in passing, in order to assess the validity of the coupling. In section 2.3, figure 1b, suggests one-way coupling only i.e. mHm provides "groundwater recharge and base flow as boundary conditions to mHm" (p 3, l 16-17). Since mHm does not include groundwater, how can the calculation of these fluxes be mechanistic (p 3, l 15), because groundwater recharge strongly depends on the dynamics of the water table? Thus, the scarce information provided in this section in combination with the statements in the introduction are misleading to the reader.

This is an important observation by the reviewer, since these sections need to contain the relevant information to enable the reader to replicate our results. To address this current shortcoming, we will expand section 2.1 and 2.2 and make the description more clear. We would like to state our basic coupling principle as the following paragraph and add the two paragraphs into the revised manuscript:

The current mHM#OGS model is a one-way coupling model and focuses on the assessment of infiltration-excess recharge (IER) and linear baseflow (LB). Considering the different equation systems of two models (ODEs in mHM and PDEs in OGS), the mechanical coupling that fully satisfy conservation of mass, energy and

momentum is theoretically impossible. The one-way coupling method can guarantee conservation of mass and was used in this study.

We will also add Figure 1 and its corresponding explanation into the revised manuscript. We believe the readers will get a clear picture of our modeling approach in the revised manuscript.

4. Section 2.3.2 with the title "Boundary condition-based coupling" provides the basic equations, yet leaves the reader wondering how the coupling is really done. Something is said about the exchange of fluxes via qe and qe' (p 7, l 3), but these are sources not boundary fluxes. What is equation 2? The upper boundary condition for the groundwater flow model? Shouldn't the coupling be performed via equation 2 as promised in the section title? In addition, the authors state that "the coupling interface converts time series of variables and fluxes to Neumann boundary conditions...". How does that fit in? This reader is left confused.

Again, this is an important observation by the reviewer. We admit that the qe and qe' in equation 1 is redundant and will confuse the readers. It is the equation 2 that works to connect mHM and OGS. We will delete the equation 1 and revise section 2.3.2 carefully to make sure the "boundary condition-based coupling" is properly presented. With regard to the sentence "the coupling interface converts time series of variables and fluxes to Neumann boundary conditions...", it means that the boundary condition-based coupling is performs by interpolating recharge and baseflow in the interface GIS2FEM, e.g., from coarser grid size in mHM to the finer grid size in OGS. In the revised manuscript, we will restructure this section following the reviewer's comments.

5. Figure 2 is not instructive. What is GIS2FEM doing? Interpolating? How does the coupling work in the vertical direction for each column? As I understand, mHm has a fixed column depth. Can the water table rise into the column along e.g. river corridors? And where does the baseflow go in OGS? How is groundwater storage in mHm (p 7, l 9-10) related to OGS? There is apparently no backward exchange with mHm due to baseflow and exchange with river networks, and no capillary rise. This reader is left confused.

We appreciate this constructive criticism. Explaining this tool appropriately is indeed necessary for the understanding of the coupling procedure and must therefore not be omitted. GIS2FEM is the model interface which is used to interpolate recharge and baseflow between different grid sizes of two models (p 7, l 19-24).

The baseflow is not determined by OGS. Instead, it is determined by the linear reservoir in mHM and then routed into the runoff (see Figure 1). The water table cannot rise into the column along river corridors because we use the linear groundwater storage in mHM to calculate baseflow. The linear reservoir is a

simplified reservoir with an overall aim of predicting runoff, whereby the dynamic interaction with groundwater head is conceptualized and simplified in order to keep the robustness of parameterization scheme, which is a unique feature of mHM.

6. On p 7, l 17-18, what do the authors mean by conversion between volumetric flux, specific flux and water head? Where in the coupling is this conversion required and why does the cell sizes need to be adjusted (there is actual re-gridding going on)?

   Thank you for your questions. The conversion is in terms of unit conversion, e.g., from distributed recharge in mHM (m/s) to volumetric recharge in OGS (m$^3$/s). There is no re-gridding going on. The boundary fluxes are directly interpolated from mHM to OGS using the interface GIS2FEM.

7. From table 2 it appears that in the author's eyes, coupling and integrated modeling of the terrestrial water cycle simply means to pass groundwater recharge values from a 1D hydrologic land surface scheme to a steady state groundwater flow model and return a head value back as some lower (boundary) condition for the hydrologic scheme (not indicated in figure 1). I feel, in the geosciences, we moved beyond this type of approach quite some time ago.

   Thank you for your comments. We are, however, afraid, that some of the reviewer's comments here are at least in part based on a misunderstanding. We will modify table 2 accordingly so that the right information can be clearly conveyed. The reviewer said "pass groundwater recharge values from a 1D hydrologic land surface scheme to a steady state groundwater flow model and return a head value back as some lower (boundary) condition for the hydrologic scheme", which is unfortunately a misunderstanding. The modeling system is basically one-way pass, which means infiltration-excess recharge (IER) and linear baseflow (LB) are calculated by mHM alone, and then passed to OGS as an upper boundary condition to force the transient groundwater model (please see Figure 1).

   To better motivate this strategy, we would like to elaborate on this decision by continuing the discussion form Comment 2. As mentioned there, we are not aiming to develop a single, seamless, mechanistic, integrated model. Instead, we are trying to establish a "hybrid model" that bridges the gap between two distinct models and makes use of the best of their abilities (see also our answers to Comment 2). These two models have different paradigms and address different challenges; First mHM, which aims for a good prediction ability of discharge across multiple time scales as well as multiple spatial-scale catchments. All of it in a computationally efficient way by using ODE's for each compartment. Second, OGS which solves computationally-expensive PDEs that directly implement flow and transport processes by using modern tools like Finite Element Method (see schematic in Figure 2). In order to

achieve a two-way coupling model, strong revisions to the implementation of these tools are necessary that will affect in particular the parametrization process of mHM.

The currently described one-way coupling can be seen as the intermediate move towards such a fully-coupled hybrid model. However, next to leading to such a more thorough coupling, the one-way coupling, described here, has a number of advantages that make it a viable modeling strategy in and of itself. First, the one-way coupling can be regarded as a safe or conservative approach, such that the parametrization process, which is one of its most salient features of mHM, remains fully intact. That way, we do not compromise any of its well-established features, such as calibration of model parameters at different scales and good runoff prediction ability, while getting in addition very good estimates of groundwater storage, flow paths and travel times. The lack of mHM to provide good estimates for these quantities has been noted in the past (see, e.g., Heße et al. 2016; Rakovec et al. 2016) and extends therefore the predictive abilities of mHM. Second, using such a one-way coupling will allow users of mHM to simply extend currently established catchment models and extend their abilities in the aforementioned way. Using a more sophisticated two-way coupling, would mean that user would have to re-establish these models almost from scratch. Third, even in the future, a one-way coupling would allow to easily expand the predictive power of a mHM catchment model if the practitioners later decide to do so, therefore leaving the option open. In short, unlike a two-way coupling, the one-way coupling described here allows the user to expand the abilities of mHM without sacrificing any of its well-known and well-established properties (Kumar et al., 2013; Rakovec et al., 2016; Samaniego et al., 2010, 2017).

In addition to improving the predictive power of mHM, OGS is gaining a strong advantage for the description of the top boundary condition, i.e., the recharge, which is temporal and spatially variable through the input of mHM. Even more, the recharge fluxes provided are based on mHM's phenomenological process description, which significantly better describes the surface level recharge fluxes than common approaches through empirical relations derived recharge rates. In the future, we additionally plan to advance in the description of water fluxes between surface and groundwater compartments through the coupled feedback between both simulation tools.To further explain the motivation for the presented one-way coupling, we like to detail some relevant research questions that can now already be answered with our model; Kumar et al (2016) have demonstrated that the Standardized Precipitation Index (SPI) has a limited applicability and low reliability in characterizing groundwater drought. Our model can be a useful tool in predicting groundwater drought & flood under different climate conditions (please check Figure 11 and 13 in the referenced manuscript). Moreover, the coupled model can be used to quantify the catchment scale legacy nitrogen stores in groundwater reservoirs. Recent research

shows that a large portion of legacy nitrogen can be older than 10 years (Van Meter et al., 2017). The current version of mHM#OGS fits well with the long-term simulation of nitrogen transport in terrestrial water cycle. The combination of process uncertainty at surface hydrology and data uncertainty at subsurface hydrology is challenging to understand travel time distributions (TTDs) at catchment scale (Benettin et al., 2015; Bertuzzo et al., 2013; Botter et al., 2010; van der Velde et al., 2015). The coupled model mHM#OGS is valuable at TTDs simulations based on the high-reputation of two modeling codes in each other's fields. In addition, field and modeling experiments at large scales suggest that the way bottom boundaries, bedrock interfaces, and other layers are treated will have a large impact on hydrological response (e.g., groundwater heads) (Broda et al., 2011; Buttle and McDonald, 2002; Ebel et al., 2008; Uchida et al., 2002, 2003).

Finally, we would like to conclude by saying that establishing a fully tow-way-coupled hybrid model, which also accounts for dynamic interaction of SW and GW, is a high priority. However, based on the challenges outlined above as well as the problem that such a model would sacrifice some of the predictive power of mHM (e.g., discharge), we consider the present coupling strategy a valuable and viable alternative in its own right, both for the meantime and the future.

To better convey these points, we will revise the introduction section of the manuscript accordingly.

8. The description of the study area and model setup, calibration etc. belong into a separate section.
The results can not be assessed unfortunately, because of the poor explanation of the applied modeling and coupling techniques.

We appreciate this observation. If other reviewers do not explicitly argue against this notion, we will separate this section into two sections in the revised version of the manuscript. There, we will also provide the source code of the coupled system, the test case along with all needed data in the Github repository in order to facilitate all interested people.

9. Language and grammar require considerable improvement.

Thank you. We will thoroughly revise the manuscript and check it with a native English speaker.

**References**

Ajami, H., McCabe, M. F., Evans, J. P. and Stisen, S.: Assessing the impact of model

spin-up on surface water-groundwater interactions using an integrated hydrologic model, Water Resour. Res., 50, 1–21, doi:10.1002/2013WR014258.Received, 2014.

Benettin, P., Kirchner, J. W., Rinaldo, A. and Botter, G.: Modeling chloride transport using travel time distributions at Plynlimon, Wales, Water Resour. Res., 3259–3276, doi:10.1002/2014WR016600, 2015.

Bertuzzo, E., Thomet, M., Botter, G. and Rinaldo, A.: Catchment-scale herbicides transport: Theory and application, Adv. Water Resour., 52, 232–242, doi:10.1016/j.advwatres.2012.11.007, 2013.

Botter, G., Bertuzzo, E. and Rinaldo, A.: Transport in the hydrologic response: Travel time distributions, soil moisture dynamics, and the old water paradox, Water Resour. Res., 46(3), 1–18, doi:10.1029/2009WR008371, 2010.

Broda, S., Paniconi, C. and Larocque, M.: Numerical investigation of leakage in sloping aquifers, J. Hydrol., 409(1), 49–61, doi:https://doi.org/10.1016/j.jhydrol.2011.07.035, 2011.

Buttle, J. M. and McDonald, D. J.: Coupled vertical and lateral preferential flow on a forested slope, Water Resour. Res., 38(5), 16–18, doi:10.1029/2001WR000773, 2002.

Camporese, M., Paniconi, C., Putti, M. and Orlandini, S.: Surface-subsurface flow modeling with path-based runoff routing, boundary condition-based coupling, and assimilation of multisource observation data, Water Resour. Res., 46(2), 2010.

Chen, X. and Hu, Q.: Groundwater influences on soil moisture and surface evaporation, J. Hydrol., 297(1), 285–300, 2004.

Cuthbert, M. O., MacKay, R. and Nimmo, J. R.: Linking soil moisture balance and source-responsive models to estimate diffuse and preferential components of groundwater recharge, Hydrol. Earth Syst. Sci., 17(3), 1003–1019, doi:10.5194/hess-17-1003-2013, 2013.

Delfs, J. O., Blumensaat, F., Wang, W., Krebs, P. and Kolditz, O.: Coupling hydrogeological with surface runoff model in a Poltva case study in Western Ukraine, Environ. Earth Sci., 65(5), 1439–1457, doi:10.1007/s12665-011-1285-4, 2012.

Ebel, B. A., Loague, K., Montgomery, D. R. and Dietrich, W. E.: Physics-based continuous simulation of long-term near-surface hydrologic response for the Coos Bay experimental catchment, Water Resour. Res., 44(7), n/a--n/a, doi:10.1029/2007WR006442, 2008.

Fang, K. and Shen, C.: Full-flow-regime storage-streamflow correlation patterns provide insights into hydrologic functioning over the continental US, Water Resour. Res., 1–20, doi:10.1002/2016WR020283, 2017.

Ferguson, I. M. and Maxwell, R. M.: Role of groundwater in watershed response and land surface feedbacks under climate change, Water Resour. Res., 46(10), n/a--n/a, doi:10.1029/2009WR008616, 2010.

De Graaf, I. E. M., Sutanudjaja, E. H., Van Beek, L. P. H. and Bierkens, M. F. P.: A high-resolution global-scale groundwater model, Hydrol. Earth Syst. Sci., 19(2), 823–837, doi:10.5194/hess-19-823-2015, 2015.

Graham, D. N. and Butts, M. B.: Flexible, integrated watershed modelling with MIKE SHE, Watershed Model., 849336090, 245–272, 2005.

Guzman, J. A., Moriasi, D. N., Gowda, P. H., Steiner, J. L., Starks, P. J., Arnold, J. G. and Srinivasan, R.: A model integration framework for linking SWAT and MODFLOW, Environ. Model. Softw., 73, 103–116, doi:10.1016/j.envsoft.2015.08.011, 2015.

Falk Heße, Matthias Zink, Rohini Kumar, Luis Samaniego, and Sabine Attinger: Spatially distributed characterization of soil-moisture dynamics using travel-time distributions. Hydrol. Earth Syst. Sci., 20, 1–22, 2016 doi:10.5194/hess-20-1-2016

Huntington, J. L. and Niswonger, R. G.: Role of surface-water and groundwater interactions on projected summertime streamflow in snow dominated regions: An integrated modeling approach, Water Resour. Res., 48(11), 1–20, doi:10.1029/2012WR012319, 2012.

Hunt, R. J., Walker, J. F., Selbig, W. R., Westenbroek, S. M. and Regan, R. S.: Simulation of Climate - Change effects on streamflow, Lake water budgets, and stream temperature using GSFLOW and SNTEMP, Trout Lake Watershed, Wisconsin, USGS Sci. Investig. Report., 2013–5159, 2013.

Hwang, H. T., Park, Y. J., Sudicky, E. A. and Forsyth, P. A.: A parallel computational framework to solve flow and transport in integrated surface-subsurface hydrologic systems, Environ. Model. Softw., 61, 39–58, doi:10.1016/j.envsoft.2014.06.024, 2014.

Ivanov, V. Y., Bras, R. L. and Vivoni, E. R.: Vegetation-hydrology dynamics in complex terrain of semiarid areas: 1. A mechanistic approach to modeling dynamic feedbacks, Water Resour. Res., 44(3), 2008.

Ivanov, V. Y., Fatichi, S., Jenerette, G. D., Espeleta, J. F., Troch, P. A. and Huxman, T. E.: Hysteresis of soil moisture spatial heterogeneity and the "homogenizing" effect of vegetation, Water Resour. Res., 46(9), n/a--n/a, doi:10.1029/2009WR008611, 2010.

Kim, N. W., Chung, I. M., Won, Y. S. and Arnold, J. G.: Development and application of the integrated SWAT--MODFLOW model, J. Hydrol., 356(1), 1–16, 2008.

Koirala, S., Yeh, P. J. F., Hirabayashi, Y., Kanae, S. and Oki, T.: Global-scale land surface hydrologic modeling with the representation of water table dynamics, J. Geophys. Res., 119(1), 75–89, doi:10.1002/2013JD020398, 2014.

Kumar, M., Duffy, C. J. and Salvage, K. M.: A second-order accurate, finite volume--based, integrated hydrologic modeling (FIHM) framework for simulation of surface and subsurface flow, Vadose Zo. J., 8(4), 873–890, 2009.

Kumar, R., Musuuza, J. L., Van Loon, A. F., Teuling, A. J., Barthel, R., Ten Broek, J., Mai, J., Samaniego, L. and Attinger, S.: Multiscale evaluation of the Standardized

Precipitation Index as a groundwater drought indicator, Hydrol. Earth Syst. Sci., 20(3), 1117–1131, doi:10.5194/hess-20-1117-2016, 2016.

Liang, X., Xie, Z. and Huang, M.: A new parameterization for surface and groundwater interactions and its impact on water budgets with the variable infiltration capacity (VIC) land surface model, J. Geophys. Res., 108(Vic), 8613–8629, doi:10.1029/2002JD003090, 2003.

Markstrom, S. L., Niswonger, R. G., Regan, R. S., Prudic, D. E. and Barlow, P. M.: GSFLOW—Coupled Ground-Water and Surface-Water Flow Model Based on the Integration of the Precipitation-Runoff Modeling System (PRMS) and the Modular Ground-Water Flow Model (MODFLOW-2005), U.S. Geol. Surv., (Techniques and Methods 6-D1), 240 [online] Available from: http://pubs.er.usgs.gov/publication/tm6D1, 2008.

Maxwell, R. M. and Miller, N. L.: Development of a coupled land surface and groundwater model, J. Hydrometeorol., 6(3), 233–247, doi:10.1175/JHM422.1, 2005.

Maxwell, R. M., Condon, L. E. and Kollet, S. J.: A high-resolution simulation of groundwater and surface water over most of the continental US with the integrated hydrologic model ParFlow v3, Geosci. Model Dev., 8(3), 923–937 [online] Available from: http://www.geosci-model-dev.net/8/923/2015/ (Accessed 3 February 2016), 2015.

Van Meter, K. J., Basu, N. B. and Van Cappellen, P.: Two centuries of nitrogen dynamics: Legacy sources and sinks in the Mississippi and Susquehanna River Basins, Global Biogeochem. Cycles, 31(1), 2–23, doi:10.1002/2016GB005498, 2017.

Rakovec, O., Kumar, R., Mai, J., Cuntz, M., Thober, S., Zink, M., Attinger, S., Schäfer, D., Schrön, M., and Samaniego, L.: Multiscale and multivariate evaluation of water fluxes and states over European river basins, J. Hydrometeorol., 17, 287–307, doi:10.1175/JHM-D-15-0054.1, 2016.

Panday, S. and Huyakorn, P. S.: A fully coupled physically-based spatially-distributed model for evaluating surface/subsurface flow, Adv. Water Resour., 27(4), 361–382, doi:https://doi.org/10.1016/j.advwatres.2004.02.016, 2004.

Paniconi, C. and Putti, M.: Physically based modeling in catchment hydrology at 50: Survey and outlook, Water Resour. Res., 51(9), 7090–7129, doi:10.1002/2015WR017780, 2015.

Phi, S., Clarke, W. and Li, L.: Laboratory and numerical investigations of hillslope soil saturation development and runoff generation over rainfall events, J. Hydrol., 493(Supplement C), 1–15, doi:https://doi.org/10.1016/j.jhydrol.2013.04.009, 2013.

Qu, Y. and Duffy, C. J.: A semidiscrete finite volume formulation for multiprocess watershed simulation, Water Resour. Res., 43(8), n/a--n/a, doi:10.1029/2006WR005752, 2007.

Rigon, R., Bertoldi, G. and Over, T. M.: GEOtop: A Distributed Hydrological Model

with Coupled Water and Energy Budgets, J. Hydrometeorol., 7(3), 371–388, doi:10.1175/JHM497.1, 2006.

Rihani, J. F., Maxwell, R. M. and Chow, F. K.: Coupling groundwater and land surface processes: Idealized simulations to identify effects of terrain and subsurface heterogeneity on land surface energy fluxes, Water Resour. Res., 46(12), 1–14, doi:10.1029/2010WR009111, 2010.

Samaniego, L., Kumar, R. and Attinger, S.: Multiscale parameter regionalization of a grid-based hydrologic model at the mesoscale, Water Resour. Res., 46(5), n/a-n/a, doi:10.1029/2008WR007327, 2010.

Samaniego, L., Kumar, R., Thober, S., Rakovec, O., Zink, M., Wanders, N., Eisner, S., Schmied, H. M., Sutanudjaja, E. H., Warrach-Sagi, K. and others: Toward seamless hydrologic predictions across spatial scales, Hydrol. Earth Syst. Sci., 21(9), 4323, 2017.

Shen, C. and Phanikumar, M. S.: A process-based, distributed hydrologic model based on a large-scale method for surface-subsurface coupling, Adv. Water Resour., 33(12), 1524–1541, doi:10.1016/j.advwatres.2010.09.002, 2010.

Smerdon, B. D., Mendoza, C. A. and Devito, K. J.: Simulations of fully coupled lake-groundwater exchange in a subhumid climate with an integrated hydrologic model, Water Resour. Res., 43(1), n/a-n/a [online] Available from: http://onlinelibrary.wiley.com/doi/10.1029/2006WR005137/full (Accessed 19 August 2016), 2007.

Spanoudaki, K., Stamou, A. I. and Nanou-Giannarou, A.: Development and verification of a 3-D integrated surface water–groundwater model, J. Hydrol., 375(3), 410–427, doi:https://doi.org/10.1016/j.jhydrol.2009.06.041, 2009.

Sutanudjaja, E. H., Van Beek, L. P. H., De Jong, S. M., Van Geer, F. C. and Bierkens, M. F. P.: Large-scale groundwater modeling using global datasets: A test case for the Rhine-Meuse basin, Hydrol. Earth Syst. Sci., 15(9), 2913–2935, doi:10.5194/hess-15-2913-2011, 2011.

Sutanudjaja, E. H., Van Beek, L. P. H., De Jong, S. M., Van Geer, F. C. and Bierkens, M. F. P.: Calibrating a large-extent high-resolution coupled groundwater-land surface model using soil moisture and discharge data, Water Resour. Res., 50(1), 687–705, doi:10.1002/2013WR013807, 2014.

Therrien, R., McLaren, R. G., Sudicky, E. A. and Panday, S. M.: HydroGeoSphere: A three-dimensional numerical model describing fully-integrated subsurface and surface flow and solute transport, Groundw. Simulations Group, Univ. Waterloo, Waterloo, 2010.

Uchida, T., Kosugi, K. and Mizuyama, T.: Effects of pipe flow and bedrock groundwater on runoff generation in a steep headwater catchment in Ashiu, central Japan, Water Resour. Res., 38(7), 14–24, doi:10.1029/2001WR000261, 2002.

Uchida, T., Asano, Y., Ohte, N. and Mizuyama, T.: Seepage area and rate of bedrock

groundwater discharge at a granitic unchanneled hillslope, Water Resour. Res., 39(1), n/a--n/a, doi:10.1029/2002WR001298, 2003.

van der Velde, Y., Heidbüchel, I., Lyon, S. W., Nyberg, L., Rodhe, A., Bishop, K. and Troch, P. A.: Consequences of mixing assumptions for time-variable travel time distributions, Hydrol. Process., 29(16), 3460–3474, doi:10.1002/hyp.10372, 2015.

VanderKwaak, J. E. and Loague, K.: Hydrologic-response simulations for the R-5 catchment with a comprehensive physics-based model, Water Resour. Res., 37(4), 999–1013, doi:10.1029/2000WR900272, 2001.

Weill, S., Mouche, E. and Patin, J.: A generalized Richards equation for surface/subsurface flow modelling, J. Hydrol., 366(1–4), 9–20, doi:10.1016/j.jhydrol.2008.12.007, 2009.

Yeh, P. J. F. and Eltahir, E. A. B.: Representation of water table dynamics in a land surface scheme. Part I: Model development, J. Clim., 18(12), 1861–1880, 2005.

---

## Referee Comment (RC2) · Anonymous Referee #2 · 13 Nov 2017

Overall:

This is a poor paper. The two models the authors have used in their catchment simulation are not described in sufficient detail to enable a reader to understand how all the processes have been implemented. Particularly lacking is how the exchange fluxes are handled. This is surprising given that the focus of the paper is on model coupling (as stated in the title). In addition, the groundwater model is incomplete, as the authors do not describe how a water table is handled in the model, specifically the role of specific yield. This is a major omission, given the influence it has on water table dynamics, a

key measure used to assess the model performance. Finally, there is no mention of river geometry (e.g. river width) and how water levels are converted into flows. Given this incompleteness, it is not possible to give any comment on the quality of the model simulations presented in the paper and the author need to address these details in any subsequent resubmission.

Specific remarks:

Eq1 refers solely to changes in pressure head being governed by the specific storage coefficient. However, this refers to changes in storage due to water and rock compressibility (see Freeze and Cherry, 1979) and, therefore, is primarily associated with storage change in confined aquifers. In an unconfined aquifer, which is the focus of the paper here, storage changes are largely governed by changes in the water table and the wetting and dewatering of pores, which is typically characterised by the specific yield. It is not clear from the description of model how this is handled. Furthermore, there is no reference to specific yield in the text and, as this is an important parameter which has a major influence on groundwater dynamics, it's omission makes commenting on the model's performance rather difficult.

Eq1. There are two fluxes $q_s$ and $q_e$ included in the groundwater continuity equation. $q_s$ is defined as a specified rate source/sink. Presumably, this refers both to abstractions of water from wells as well as recharge from rainfall infiltration in contrast to the flux $q_e$, which is defined as the exchange with surface water. Furthermore, in Eq3. the surface water continuity equation, a flux $q_e^{'}$ is referred to as the exchange rate with surface water. It is not clear to me what are the differences between these two terms, mainly because, in both cases, no details are given on how these fluxes are calculated. This is particularly problematic, as a key feature of the paper (and referred to in the title) is the coupling between the surface and subsurface models. I would, therefore, have expected to see an equation that includes both $\psi_p$ and $\psi_s$ showing how the models are explicitly coupled.

The authors cite Camporese et al. (2010) in their discussion on the two coupling terms, however, there are some important differences. Camporese at al. (2010) do not appear to have an equivalent flux for $q_e$. They have a term in their surface water balance equation that looks to be the equivalent of $q_e^{'}$ (which they refer to as $q_s$), however, even here the exact definition is not given. Furthermore, Camporese et al. (2010) solve Richards' equation, rather than the saturated groundwater flow equation, in their subsurface model and, therefore, where the water table is below the base of the river, the coupling would be completely different.

Finally, in connection with the surface-subsurface coupling, there isn't any reference to river geometry and its role in calculating exchange fluxes and river flow (e.g. as shown in Fig 10).

Typographic errors:

Eq1. Note z as specified here denotes depth. The description of vertical coordinate is not clear.

Eq. 2, the pressure term, should have a p subscript.

Reference:

Freeze, R.A., and Cherry, J.A., 1979, Groundwater: Englewood Cliffs, NJ, Prentice-Hall, 604 p.

---

## Referee Comment (RC3) · E.H. Sutanudjaja (Referee) · 16 Nov 2017

Review on "Improved representation of groundwater at a regional scale – coupling of mesocale Hydrologic Model (mHM) with OpenGeoSys"

General comment:

This paper deals with an effort to couple the regional scale mHM model to a groundwater flow model (i.e. OGS) that can simulate groundwater lateral flow and groundwater head dynamics.

This topic fits very well to the scope of this journal and I consider this study is an im-

portant contribution for regional or large scale hydrological modelling efforts. Currently, there are a still quite limited number of regional (large) scale hydrological models that include lateral groundwater flow component and can simulate groundwater head dynamics. An extension to groundwater head simulation will greatly strengthen the mHM model capabilities, e.g. for enhancing their groundwater drought studies and groundwater transport modelling.

As a test case, the authors used the Naegelstedt catchment where head observation data are available. They managed to show some convincing validation results of their groundwater modelling result to observation data (e.g. Figure 11). The authors deserved credit for their extensive and successful modelling experiment.

However, this paper is still poorly written and therefore it is difficult to comprehend. English must be improved. I strongly recommend that the revised version is checked by an English native speaker.

Below, I provided a (non-exhaustive) list of some remarks and suggestions that can be used to revise the manuscript.

Details / specific comments:

Page 1, lines 1-2: I suggest to rephrase this sentence. Most hydrological models do include groundwater component, e.g including groundwater (vertical) recharge component and using a linear reservoir concept for groundwater baseflow/discharge. Yet, they hardly include lateral groundwater flow component and simulate groundwater head dynamics.

Page 1, lines 8-9: The sentence (Nested time stepping . . .) does not really flow with the previous ones. Please rephrase. - It will be very informative if the time step lengths used (for both models) are mentioned in the abstract. If I understand correctly, the time step length used for mHM was daily, while OGS used monthly time step. Am I correct?

Page 1, lines 15-16: Please clarify with what you meant by the 'offline coupling method'

in your study.

Page 1, lines 15-16: How much is the 'little surplus' in your computational cost?

Page 2, line 8: ... ignoring lateral groundwater flow ...

Page 2, lines 32-35 and page 3, lines 1-10: Please rewrite this paragraph. I found its sentences (e.g. the first until fourth sentences) do not really flow and connect with each other.

Page 3, line 4: LSM? Common Land Model? I guess that you meant CLM (Community Land Model).

Page 3, line 6: For this study, were you using a similar offline coupling strategy as used by Sutanudjaja et al. (2011). Did you first run the mHM model for the entire model simulation period (1970-2005?), then use the mHM output to force the OGS model? Please clarify.

Page 3, line 8: GSFLOW? What does GSFLOW stand for?

Page 3, line 14: What is THMC? I cannot find its long form of this acronym before this line.

Page 3, lines 15-17: Please rephrase this sentence. I am not sure what you meant by 'offline' coupled here.

Page 3, line 17: ... an offline coupled model ...

Page 4, line 8: So, did you apply MPR for the current study? This is not really clear for me.

Page 4, line 28: ... first and second regions ...

Page 4, lines 31-32: Could you please elaborate with what you meant by 'sequential boundary condition switching technique'?

Figure 1: I cannot find the explanations for GOCAD, GO20GS and PEST in the

text/paragraph.

Page 5, line GIS2FEM: What does GIS2FEM stand for?

Page 6, lines 7-15: Could you please check this part. I guess that there are some missing lines or sentence here. For example, I cannot find the introduction and explanation for Eq. 2.

Page 7, lines 9-10: Due to this liner reservoir conceptualization, I guess that the current coupled model mHM#OGS cannot simulate infiltration from surface water bodies (rivers) to groundwater?

Page 9, line 24: What is VTU?

Section 2.5: Please rewrite this section, particularly to clarify/confirm the following: - So, you have two scenarios of groundwater modelling: SC1: spatially distributed recharge and SC2: homogeneous recharge - Did you calibrate both scenarios groundwater modelling independently? Or, did you just calibrate SC1 and then using the calibrated SC1 parameters for SC2?

Page 16, line 2: Please provide the unit (m2?) for 8625 and 464.74.

Page 16, lines 2-3: What do you mean by the calibration is robust with totally 114 model runs?

Page 16, lines 3-4: What do you mean by 'convergence criteria relevant to observation'? Please rephrase the sentence.

Page 16, lines 14-16: The sentence does not flow with the previous ones.

Figure 9: I guess this map is for a steady-state condition. Please clarify.

Page 16, line 19-20: What do you mean by the last sentence, i.e. the coincidence with Wechsung (2005)? Is it possible to include/visualize some figures from Wechsung (2005)?

Figure 10: Could you please also provide other performance metrics, e.g. NSE, KGE?

I missed some crucial information, such as the resolution of the forcing data used and the resolution of mHM model used.

Page 19, line 5: . . . each monitoring well ... (singular)

Page 19, lines 15-17, Page 20: Please check the English. An example: Another reason is that we assigned a homogenous storage coefficient (?) in all aquifers, which an over-simplified setting.

Page 21, line 14: Did Kumar et al. (2016) also simulate groundwater heads?

Page 22, lines 9-17: For prediction/application in ungauged basins, I believe that hydrogeological characterization (in ungauged basins) still remains as one of the main challenges.

––––––––––––––––––––

---

## Author Response (AR1)

**Responses to Anonymous Referee 1**

We thank the referee reviewer very much for his comprehensive and insightful comments. Those comments are really helpful for us to revise the manuscript.

Before we reply to any specific questions of the comments, we would like to clarify two points.

The first point is that we would like to explain the linear groundwater reservoir in mHM, which was not directly explained in the manuscript since it has been included in the references (Kumar et al., 2013; Samaniego et al., 2010). mHM contains a linear reservoir to generate daily baseflow (please see Figure 1). The generated baseflow of each grid are further routed into streams using Muskingum-Cunge method. In the coupled model mHM#OGS, we take spatially distributed recharge and routed baseflow generated by the linear reservoir, then feed these two boundary sources to GIS2FEM (the coupling interface to convert unit and adjust time step), and then to OGS as upper boundary conditions. The baseflow is still calculated by the linear reservoir in mHM and routed into runoff (please see Figure 1). We have now noticed that the detailed explanation of the linear groundwater reservoir is essential and will include it into the revised manuscript.

The second point is that we are not aiming to develop a fully physically-based model. We are not aiming to study the mechanistic interaction of soil-zone processes and the groundwater heads. Instead, we are aiming to develop an open-source regional-scale model which can predict catchment runoff and groundwater head dynamics simultaneously, while preserves all existing and well-tested mHM features, e.g., the parameterization scheme (Kumar et al., 2013; Rakovec et al., 2016; Samaniego et al., 2010, 2017).

[Figure]

a) Original structure of mHM        b) Structure of mHM#OGS

Figure 1 mHM#OGS as an approach into realization of spatially distributed groundwater head.

**Major comments**

1. The authors present a coupling approach for a land surface hydrologic and ground water flow model, mHM and OGS respectively. The manuscript contains sections on the coupling, model setup over a real catchment and verification of the results. The model coupling is not explained appropriately and it's not clear, whether the coupling approach satisfies the current state-of-the-art published in GMD. Based on the provided explanation, the results cannot be assessed unfortunately.

   Thank you for your comment. Enabling the reader to independently reproduce the results is an important aspect of the publishing process of GMD. To improve that part, we will significantly revise the model section in order to make our approach more clear to the reader and avoid misconception on our work. We will also provide a fully accessible code, a test example together with all needed data in the Github repository.

2. Introduction The introduction is incomplete and misses some of the most important and heavily cited references of integrated models and modeling studies of the terrestrial water cycle. Apparently the authors are not aware of the state-of-the-art. Proper citation of the mentioned models is missing. Is the sole goal of the introduction to promote the work of the co-authors (e.g. statement p 3, l 12-15 and citations throughout)?

Thank you for your insights. We will revise the whole introduction section accordingly and cite all the up-to-date papers properly.

To better convey these points and avoid possible future misunderstanding, we will revise the introduction section in manuscript accordingly. In addition, we further expand our literature review by properly referencing integrated surface/subsurface hydrologic models (ISSHMs) such as InHM (Smerdon et al., 2007; VanderKwaak and Loague, 2001), Parflow (Maxwell et al., 2015), tRIBS ((Ivanov et al., 2004), CATHY (Camporese et al., 2010), GSFLOW (Hunt et al., 2013; Markstrom et al., 2008), HydroGeoSphere (Hwang et al., 2014; Therrien et al., 2010), MIKE SHE (Graham and Butts, 2005), MODHMS (Panday and Huyakorn, 2004; Phi et al., 2013), GEOtop (Rigon et al., 2006), IRENE (Spanoudaki et al., 2009), CAST3M (Weill et al., 2009), PIHM (Kumar et al., 2009; Qu and Duffy, 2007) and PAWS(Shen and Phanikumar, 2010), in the revised manuscript. The coupled land surface / groundwater models (CLSGMs) include ParFlow-CLM (Ferguson and Maxwell, 2010; Maxwell and Kollet, 2008; Maxwell and Miller, 2005; Rihani et al., 2010), tRIBS + VEGGIE (Ivanov et al., 2008, 2010), SWAT and MODFLOW (Guzman et al., 2015; Kim et al., 2008), PCR-GLOBWB-MOD (Sutanudjaja et al., 2014), SWMM-OGS (Delfs et al., 2012). Nevertheless, we will revise this section in order to convey comprehensive information of the state-of-the-art science.

Next, we would like to clarify that the within the context of our manuscript a "coupled model" is not the same as a "physically-based" or "mechanistic" integrated model. We will include this point into the storyline in the revised manuscript. What we want to develop is a hybrid model that is using two different modeling paradigms which can be easily applied in regional and continental scale, rather than a mechanistic integrated model. Our reasons for this decision is that more conceptual process-based models like mHM or Noah-MP are good at predicting quantities like discharge but are highly conceptualized and there suffering from interpretability of certain processes (e.g., base flow and interflow components). More mechanistic models like Parflow and HydroGeoSphere are highly interpretable but show consistently worse performance when predicting runoff (Paniconi and Putti, 2015). To the best of our knowledge, the skill of simulating groundwater head dynamics at regional scale of mechanistic models are always neglected and seldom assessed by the data (e.g. GW head, tracer). At the larger scale, the assessment of modeled groundwater heads dynamics can only be found in very few publications (De Graaf et al., 2015; Sutanudjaja et al., 2011).

[Figure]

Figure 2 Different questions and challenges in surface and subsurface hydrology.

The above mentioned different abilities of more phenomenological models (e.g., mHM, Noah-MP, etc) vs. the more mechanistic models (e.g., Parflow, Hydrogeosphere, etc.) is caused by the different challenges that are posed by the different compartments of the terrestrial water cycle. One of the main challenges in the surface & near-surface storage is process uncertainty, with the fact that processes like ET, land use, land cover, snow pack, etc. are extremely complex. The process uncertainty decreases as it goes deeper and deeper into the subsurface storage. In subsurface storage, hydrological processes are under Darcy's law and therefore conceptually simpler. Meanwhile, the data uncertainty becomes more significant in deep subsurface storage than in shallow storage (see schematic in Figure 2). Therefore, proper conceptualization is needed in the shallow storage in order to deal with this process uncertainty (please see schematic in Figure 2). Owing to this point, mHM was developed as a bucket-type model to better deal with this process uncertainty by optimally leveraging the information content in the discharge data. On the other hand, OGS is a mechanistic model, i.e., it has a very low process uncertainty but large amount of data uncertainty. It is therefore optimally suited to model processes in the deeper subsurface. To use the strengths and weaknesses of both these modeling concepts, we decided to separate our modeling domain into these two compartments, a strategy that is very common in hydrology (Benettin et al., 2015; Bertuzzo et al., 2013; Botter et al., 2010; van der Velde et al., 2015), and use this different modeling paradigms for each compartment.

We will also add the following two paragraphs into the revised manuscript:

At the larger, i.e., regional scale, most of the mechanistic integrated models are based on a continuity of pressure and flux on the SW/GW interface, while the momentum balance condition is always missing (Paniconi and Putti, 2015). The runoff is generally normalized as "storage-dependent runoff" by solving Richards equation, and the grid-wise generated runoffs are routed by a routing algorithm. These models

can principally simulate the dynamic interaction of different processes with SW/GW components, e.g., the interaction of soil moisture and GW head (Cuthbert et al., 2013; Maxwell and Miller, 2005; Rihani et al., 2010; Sutanudjaja et al., 2014), the storage-runoff correlation (Fang and Shen, 2017; Huntington and Niswonger, 2012; Koirala et al., 2014; Liang et al., 2003; VanderKwaak and Loague, 2001), and the dynamical interaction between ET and GW head (Chen and Hu, 2004; Koirala et al., 2014; Yeh and Eltahir, 2005).

In constract to that, in this study, we present a one-way coupling model mHM#OGS and focuses on the representation of Infiltration-Excess Recharge (IER) and Linear Baseflow (LB) through a case study of a mesoscale catchment. The basic scientific question we want to answer is: Can spatially distributed groundwater heads and their dynamics be reasonably captured by expanding on the abilities of a phenomenological model like mHM at the regional scale? Based on the case study, we would conclude that this expansion was successful since in addition to predicting discharge, our coupled model is also able to predict head measurements as well. Since our focus is the predictive accuracy of mHM (compared to interpretability and inference), we consider the physical plausibility of the coupling of recharge and baseflow to be a means to that end and not an end in itself. Improving the plausibility of these processes will, if done right, also lead to higher predictive power. We will elaborate on these points on more details in our answers to Comment 7.

Please check the updated Introduction section in the revised manuscript.

3. Model description Section 2.1 and 2.2 must be expanded. At least, the reader must get some idea about the basic principles that are used to model the different processes mentioned in passing, in order to assess the validity of the coupling. In section 2.3, figure 1b, suggests one-way coupling only i.e. mHm provides "groundwater recharge and base flow as boundary conditions to mHm" (p 3, l 16-17). Since mHm does not include groundwater, how can the calculation of these fluxes be mechanistic (p 3, l 15), because groundwater recharge strongly depends on the dynamics of the water table? Thus, the scarce information provided in this section in combination with the statements in the introduction are misleading to the reader.

This is an important observation by the reviewer, since these sections need to contain the relevant information to enable the reader to replicate our results. To address this current shortcoming, we will expand section 2.1 and 2.2 and make the description more clear. We would like to state our basic coupling principle as the following paragraph and add the two paragraphs into the revised manuscript:

The current mHM#OGS model is a one-way coupling model and focuses on the assessment of infiltration-excess recharge (IER) and linear baseflow (LB).

Considering the different equation systems of two models (ODEs in mHM and PDEs in OGS), the mechanical coupling that fully satisfy conservation of mass, energy and momentum is theoretically impossible. The one-way coupling method can guarantee conservation of mass and was used in this study.

We will also add Figure 1 and its corresponding explanation into the revised manuscript. We believe the readers will get a clear picture of our modeling approach in the revised manuscript.

Please check the updated Section 2.1 and 2.2 in the revised manuscript.

4. Section 2.3.2 with the title "Boundary condition-based coupling" provides the basic equations, yet leaves the reader wondering how the coupling is really done. Something is said about the exchange of fluxes via qe and qe' (p 7, l 3), but these are sources not boundary fluxes. What is equation 2? The upper boundary condition for the groundwater flow model? Shouldn't the coupling be performed via equation 2 as promised in the section title? In addition, the authors state that "the coupling interface converts time series of variables and fluxes to Neumann boundary conditions...". How does that fit in? This reader is left confused.

Again, this is an important observation by the reviewer. We admit that the qe and qe' in equation 1 is redundant and will confuse the readers. It is the equation 2 that works to connect mHM and OGS. We will delete the equation 1 and revise section 2.3.2 carefully to make sure the "boundary condition-based coupling" is properly presented. With regard to the sentence "the coupling interface converts time series of variables and fluxes to Neumann boundary conditions...", it means that the boundary condition-based coupling is performs by interpolating recharge and baseflow in the interface GIS2FEM, e.g., from coarser grid size in mHM to the finer grid size in OGS.

Please check the new Section 2.3 in the revised manuscript.

5. Figure 2 is not instructive. What is GIS2FEM doing? Interpolating? How does the coupling work in the vertical direction for each column? As I understand, mHm has a fixed column depth. Can the water table rise into the column along e.g. river corridors? And where does the baseflow go in OGS? How is groundwater storage in mHm (p 7, l 9-10) related to OGS? There is apparently no backward exchange with mHm due to baseflow and exchange with river networks, and no capillary rise. This reader is left confused.

We appreciate this constructive criticism. Explaining this tool appropriately is indeed necessary for the understanding of the coupling procedure and must therefore not be omitted. GIS2FEM is the model interface which is used to interpolate recharge and baseflow between different grid sizes of two models (p 7, l 19-24).

The baseflow is not determined by OGS. Instead, it is determined by the linear reservoir in mHM and then routed into the streams (see Figure 1). The water table cannot rise into the column along river corridors because we use the linear groundwater storage in mHM to calculate baseflow. The linear reservoir is a simplified reservoir with an overall aim of predicting runoff, whereby the dynamic interaction with groundwater head is conceptualized and simplified in order to keep the robustness of parameterization scheme, which is a unique feature of mHM.

Please check the updated Figure 2 in the revised manuscript.

6. On p 7, l 17-18, what do the authors mean by conversion between volumetric flux, specific flux and water head? Where in the coupling is this conversion required and why does the cell sizes need to be adjusted (there is actual re-gridding going on)?

Thank you for your questions. The conversion is in terms of unit conversion, e.g., from distributed recharge in mHM (m/s) to volumetric recharge in OGS ($m^3$/s). There is no re-gridding going on. The boundary fluxes are directly interpolated from mHM to OGS using the interface GIS2FEM.

7. From table 2 it appears that in the author's eyes, coupling and integrated modeling of the terrestrial water cycle simply means to pass groundwater recharge values from a 1D hydrologic land surface scheme to a steady state groundwater flow model and return a head value back as some lower (boundary) condition for the hydrologic scheme (not indicated in figure 1). I feel, in the geosciences, we moved beyond this type of approach quite some time ago.

Thank you for your comments. We are, however, afraid, that some of the reviewer's comments here are at least in part based on a misunderstanding. We will modify table 2 accordingly so that the right information can be clearly conveyed. The reviewer said "pass groundwater recharge values from a 1D hydrologic land surface scheme to a steady state groundwater flow model and return a head value back as some lower (boundary) condition for the hydrologic scheme", which is unfortunately a misunderstanding. The modeling system is basically one-way pass, which means infiltration-excess recharge (IER) and linear baseflow (LB) are calculated by mHM alone, and then passed to OGS as an upper boundary condition to force the transient groundwater model (please see Figure 1).

To better motivate this strategy, we would like to elaborate on this decision by continuing the discussion form Comment 2. As mentioned there, we are not aiming to develop a single, seamless, mechanistic, integrated model. Instead, we are trying to establish a "hybrid model" that bridges the gap between two distinct models and makes use of the best of their abilities (see also our answers to Comment 2). These

two models have different paradigms and address different challenges; First mHM, which aims for a good prediction ability of discharge across multiple time scales as well as multiple spatial-scale catchments. All of it in a computationally efficient way by using ODE's for each compartment. Second, OGS which solves computationally-expensive PDEs that directly implement flow and transport processes by using modern tools like Finite Element Method (see schematic in Figure 2). In order to achieve a two-way coupling model, strong revisions to the implementation of these tools are necessary that will affect in particular the parametrization process of mHM.

The currently described one-way coupling can be seen as the intermediate move towards such a fully-coupled hybrid model. However, next to leading to such a more thorough coupling, the one-way coupling, described here, has a number of advantages that make it a viable modeling strategy in and of itself. First, the one-way coupling can be regarded as a safe or conservative approach, such that the parametrization process, which is one of its most salient features of mHM, remains fully intact. That way, we do not compromise any of its well-established features, such as calibration of model parameters at different scales and good runoff prediction ability, while getting in addition very good estimates of groundwater storage, flow paths and travel times. The lack of mHM to provide good estimates for these quantities has been noted in the past (see, e.g., Heße et al. 2016; Rakovec et al. 2016) and extends therefore the predictive abilities of mHM. Second, using such a one-way coupling will allow users of mHM to simply extend currently established catchment models and extend their abilities in the aforementioned way. Using a more sophisticated two-way coupling, would mean that user would have to re-establish these models almost from scratch. Third, even in the future, a one-way coupling would allow to easily expand the predictive power of a mHM catchment model if the practitioners later decide to do so, therefore leaving the option open. In short, unlike a two-way coupling, the one-way coupling described here allows the user to expand the abilities of mHM without sacrificing any of its well-known and well-established properties (Kumar et al., 2013; Rakovec et al., 2016; Samaniego et al., 2010, 2017).

In addition to improving the predictive power of mHM, OGS is gaining a strong advantage for the description of the top boundary condition, i.e., the recharge, which is temporal and spatially variable through the input of mHM. Even more, the recharge fluxes provided are based on mHM's phenomenological process description, which significantly better describes the surface level recharge fluxes than common approaches through empirical relations derived recharge rates. In the future, we additionally plan to advance in the description of water fluxes between surface and groundwater compartments through the coupled feedback between both simulation tools.To further explain the motivation for the presented one-way coupling, we like to detail some relevant research questions that can now already be answered with our

model; Kumar et al (2016) have demonstrated that the Standardized Precipitation Index (SPI) has a limited applicability and low reliability in characterizing groundwater drought. Our model can be a useful tool in predicting groundwater drought & flood under different climate conditions (please check Figure 11 and 13 in the referenced manuscript). Moreover, the coupled model can be used to quantify the catchment scale legacy nitrogen stores in groundwater reservoirs. Recent research shows that a large portion of legacy nitrogen can be older than 10 years (Van Meter et al., 2017). The current version of mHM#OGS fits well with the long-term simulation of nitrogen transport in terrestrial water cycle. The combination of process uncertainty at surface hydrology and data uncertainty at subsurface hydrology is challenging to understand travel time distributions (TTDs) at catchment scale (Benettin et al., 2015; Bertuzzo et al., 2013; Botter et al., 2010; van der Velde et al., 2015). The coupled model mHM#OGS is valuable at TTDs simulations based on the high-reputation of two modeling codes in each other's fields. In addition, field and modeling experiments at large scales suggest that the way bottom boundaries, bedrock interfaces, and other layers are treated will have a large impact on hydrological response (e.g., groundwater heads) (Broda et al., 2011; Buttle and McDonald, 2002; Ebel et al., 2008; Uchida et al., 2002, 2003).

Finally, we would like to conclude by saying that establishing a fully tow-way-coupled hybrid model, which also accounts for dynamic interaction of SW and GW, is a high priority. However, based on the challenges outlined above as well as the problem that such a model would sacrifice some of the predictive power of mHM (e.g., discharge), we consider the present coupling strategy a valuable and viable alternative in its own right, both for the meantime and the future.

We have revised the introduction section of the manuscript accordingly. Please check the illustration of coupling mechanism in page 7, line 12-15 and page8, line 1-23 of the revised manuscript.

8. The description of the study area and model setup, calibration etc. belong into a separate section.
The results can not be assessed unfortunately, because of the poor explanation of the applied modeling and coupling techniques.

We appreciate this observation. We have followed the reviewer's suggestion and separate this section into two sections in the revised version of the manuscript. We also provided the source code of the coupled system, the test case along with all needed data in the Github repository in order to facilitate all interested people.

9. Language and grammar require considerable improvement.

Thank you. We have thoroughly revised the manuscript and checked it with a native English speaker.

Eq 1 and Eq 3 are governing equations of surface water flow and subsurface flow, respectively.  The coupling procedures are illustrated as below:

First, the mHM grid cells are artificially classified as soil grid cells (please see brown part in Figure 4) and river grid cells (please see blue part in Figure 4). The classification method was also illustrated in the Section 2.4.3 of manuscript. $q_s$, which represents recharge in the manuscript, is calculated by the water balance equation by removing fast interflow, slow interflow and evapotranspiration from

precipitation. Meanwhile, $q_e$ is calculated only at river grid cells. At river grid cells, $q_e$ is calculated as follows:

First, baseflow is generated at every grid cell by a water balance equation combined with a linear groundwater reservoir. The released baseflow by linear groundwater reservoir (please see the detailed description of linear groundwater reservoir in next section) is then routed into the total runoff by means of a Muskingum-Cunge algorithm. The total amount of routed baseflow is then uniformly distributed to every river grid cells. The flux $q_e$, is equal to the uniformly distributed routed baseflow in each river grid cells.

Using the above scheme, the total water balance is closed because the total amount of groundwater recharge is equal to the total amount of routed baseflow.

The Eq 3 is approximated by using a Muskingum-Cunge method:

$$Q_i^1(t) = Q_i^1(t-1) + c_1 \left( Q_i^0(t-1) - Q_i^1(t-1) \right) + c_2 \left( Q_i^0(t) - Q_i^0(t-1) \right)$$

with

$$Q_i^0(t) = Q_{i'}(t) + Q_{i'}^1(t)$$

$$c_1 = \frac{\Delta t}{\kappa(1-\xi) + \frac{\Delta t}{2}}$$

$$c_2 = \frac{\frac{\Delta t}{2} - \kappa\xi}{\kappa(1-\xi) + \frac{\Delta t}{2}}$$

where

$Q_i^0$ and $Q_i^1$ denote the discharge entering and leaving the river reach located on cell i respectively.
$Q_{i'}$ is the contribution from the upstream cell $i'$.
$\kappa$ is Muskingum travel time parameter.
$\xi$ is Muskingum attenuation parameter.
$\Delta t$ is time interval.
$t$ Time index for each $\Delta t$ interval.

The Muskingum parameters, and , are calibrated by matching the historical runoff. To address this specific comment, we will add Muskingum-Cunge equation after Eq 3 as supplementary information.

Please check the revised Section 2.1 and 2.2.

[Figure]

Figure 3 mHM#OGS as an approach into realization of groundwater head dynamics

4. The authors cite Camporese et al. (2010) in their discussion on the two coupling terms, however, there are some important differences. Camporese at al. (2010) do not appear to have an equivalent flux for qe. They have a term in their surface water balance equation that looks to be the equivalent of qe 0 (which they refer to as qs), however, even here the exact definition is not given. Furthermore, Camporese et al. (2010) solve Richards' equation, rather than the saturated groundwater flow equation, in their subsurface model and, therefore, where the water table is below the base of the river, the coupling would be completely different.

As mentioned in the former paragraph, we use boundary condition-based coupling by feeding recharge to the soil grid cells (please see brown part in Figure 4), or feeding distributed routed baseflow plus recharge for the river grid cells (please see blue part in Figure 4). The calculation of distributed routed baseflow has been described in the above section. We would like to add some additional information on linear groundwater reservoir in mHM.

mHM contains a linear reservoir to generate daily baseflow (please see Figure 3**Error! Reference source not found.**). The generated baseflow of each grid are

further routed into streams using the Muskingum-Cunge method. In the coupled model mHM#OGS, we take spatially distributed recharge and routed baseflow generated by the linear reservoir, then feed these two boundary sources to GIS2FEM (the coupling interface to convert unit and adjust time step), and then to OGS as upper boundary conditions (see also our answer to Reviewer 1). The baseflow is still calculated by the linear reservoir in mHM and routed into runoff (please see Figure 1).

We have now noticed that the detailed explanation of the linear groundwater reservoir is essential, and have depicted it in the revised manuscript. Please check the Section 2.3 of the revised manuscript.

[Figure]

Figure 4 the river grids and soil grids where boundary fluxes are different

5. Finally, in connection with the surface-subsurface coupling, there isn't any reference to river geometry and its role in calculating exchange fluxes and river flow (e.g. as shown in Fig 10).

We appreciate this constructive criticism. To address it, we will add a detailed description of river geometry and its role in exchanging fluxes. The baseflow in Fig 10 is determined by mHM by routing the baseflow generated from linear reservoir of each grid (please see schematic in Figure 1).

The river geometry is displayed in Figure 5 (c) of the manuscript. Within the coupling scheme used in our study, the depth of the river, is irrelevant since we calculate the baseflow directly using mHM's inherent runoff generation and routing scheme (please see the schematic in Figure 3). From the mechanism of mHM as a grid-based hydrologic model, the river network is extracted directly from mHM grid

cells and interpolated into OGS upper surface using GIS2FEM, which is the coupling interface. Within that scheme, the width of a river is conceptually equivalent to the width of OGS grid, which is a structured grid with a width of 250 m in the manuscript. The baseflow rate is directly interpolated into surface of OGS mesh and serves as upper boundary condition of the groundwater flow model. The baseflow rate is relevant to the mHM grid size rather than OGS grid size, thus the river geometry, which is mapped in OGS mesh upper surface, has only a minor influence on the catchment scale groundwater dynamics. Since our study focuses on the catchment scale groundwater dynamics rather than the near-field groundwater flow of rivers, the coarse resolution of river network is a simple, however, efficient and reasonable setting. Nevertheless, we can also use an alternative method to portray river network, which is by defining a set of polylines in OGS geometry file. The baseflow rate is then assigned to every node within the polyline by means of linear interpolation. Using this method, the geometry of river can be better represented.

In the revised manuscript, we have incorporated the illustration of river geometry (page 11, line 19-23 and page 12, line 1-2) and modified this section accordingly.

6. Typographic errors:

   Eq1. Note z as specified here denotes depth. The description of vertical coordinate is not clear.

   Eq. 2, the pressure term, should have a p subscript.

   We will take care of these typographic errors and correct them accordingly. Thanks so much.

**Responses to Referee Review posted by E.H. Sutanudjaja**

We thank the referee reviewer very much for his comprehensive and insightful comments. Those comments are really helpful for us to revise the manuscript.

Below you could find the point-by-point response.

Major comments

10. This paper deals with an effort to couple the regional scale mHM model to a groundwater flow model (i.e. OGS) that can simulate groundwater lateral flow and groundwater head dynamics.
This topic fits very well to the scope of this journal and I consider this study is an important contribution for regional or large scale hydrological modelling efforts. Currently, there are a still quite limited number of regional (large) scale hydrological models that include lateral groundwater flow component and can simulate groundwater head dynamics. An extension to groundwater head simulation will greatly strengthen the mHM model capabilities, e.g. for enhancing their groundwater drought studies and groundwater transport modelling.
As a test case, the authors used the Naegelstedt catchment where head observation data are available. They managed to show some convincing validation results of their groundwater modelling result to observation data (e.g. Figure 11). The authors deserved credit for their extensive and successful modelling experiment.

We appreciate the reviewer's recognition of our manuscript. The reviewer pointed out that "there are a still quite limited number of regional (large) scale hydrological models that include lateral groundwater flow component and can simulate groundwater head dynamics.", which is exactly what we would like to address in the manuscript.

11. However, this paper is still poorly written and therefore it is difficult to comprehend. English must be improved. I strongly recommend that the revised version is checked by an English native speaker.

Thank you for the suggestion. We have followed the reviewer's suggestions and revised the manuscript carefully to make sure all English expressions are correct. We checked the revised manuscript carefully to make sure the English syntaxes are correct.

12. Details / specific comments:

Page 1, lines 1-2: I suggest to rephrase this sentence. Most hydrological models do include groundwater component, e.g including groundwater (vertical) recharge component and using a linear reservoir concept for groundwater baseflow/discharge. Yet, they hardly include lateral groundwater flow component and simulate groundwater head dynamics.

Thank you for this comment. We have rephrased this sentence accordingly.

13. Page 1, lines 8-9: The sentence (Nested time stepping : : :) does not really flow with the previous ones. Please rephrase. - It will be very informative if the time step lengths used (for both models) are mentioned in the abstract. If I understand correctly, the time step length used for mHM was daily, while OGS used monthly time step. Am I correct?

This is a good suggestion. Yes, the time step length used for mHM was daily, while OGS used monthly time step. We have rewritten this sentence according to the reviewer's advice. Please see page 1, line 10-11 in the revised manuscript.

14. Page 1, lines 15-16: Please clarify with what you meant by the ' offline coupling method' in your study.

To avoid any misunderstanding of the phrase "offline coupling method", we replaced the term "offline coupling method" by "one way coupling". For the details of the one way coupling scheme used in the manuscript, please check Section 2.3 in the revised manuscript.

15. Page 1, lines 15-16: How much is the ' little surplus' in your computational cost?

For each monthly groundwater time step, the simulation time is about 200s. This means the total surplus of the simulation is about 10 hour.

16. Page 2, line 8: : : : ignoring lateral groundwater flow : : :
Page 2, lines 32-35 and page 3, lines 1-10: Please rewrite this paragraph. I found its sentences (e.g. the first until fourth sentences) do not really flow and connect with each other.

We have followed the reviewer's suggestions and rewrote the paragraph accordingly. For the details, please check page 2, line 30-35 and page 2, line 1-18.

17. Page 3, line 4: LSM? Common Land Model? I guess that you meant CLM (Community Land Model).

Thank you for pointing out our misspelling. We have modified the LSM to CLM accordingly.

18. Page 3, line 6: For this study, were you using a similar offline coupling strategy as

used by Sutanudjaja et al. (2011). Did you first run the mHM model for the entire model simulation period (1970-2005?), then use the mHM output to force the OGS model? Please clarify.

We do use a similar coupling strategy as used by Sutanudjaja et al. (2011). We run the coupled model following a four-step procedure:

1) mHM is run independent of OGS to calculate land surface fluxes.
2) After mHM run was finished, the step-wise routed baseflow estimated by mHM are transformed to distributed baseflow along OGS stream network.
3) The distributed groundwater recharge generated from mHM are fed to the coupling interface GIS2FEM, and further transferred to the upper surface boundary conditions of the OGS model.
4) After mHM generated recharge and baseflow were successfully transferred to OGS upper surface boundary conditions, the groundwater model will run subsequently to simulate groundwater flow and transport processes.

Please check page7, line 11-15, and page 8, line 1-24 in the revised manuscript.

19. Page 3, line 8: GSFLOW? What does GSFLOW stand for?

GSFLOW stands for "Coupled Ground-Water and Surface-Water Flow Model Based on the Integration of the Precipitation-Runoff Modeling System (PRMS) and the Modular Ground-Water Flow Model (MODFLOW-2005)" (Markstrom et al., 2008).

20. Page 3, line 14: What is THMC? I cannot find its long form of this acronym before this line.

THMC is the short form of "Thermo-Hydro-Mechanical-Chemical (coupling)". THMC modeling is critical in many topics in the field of hydrogeology such as pollutant transport, geothermal heat exchange and seawater intrusion (Kolditz et al., 2012). OpenGeoSys (OGS) project is a scientific open-source initiative for numerical simulation of thermo-hydro-mechanical- chemical (THMC) processes in porous media.

Please see page 6, line 10 of the revised manuscript.

21. Page 3, lines 15-17: Please rephrase this sentence. I am not sure what you meant by 'offline' coupled here.
Page 3, line 17: : : : an offline coupled model ...

We have followed the reviewer's comments and rephrased this sentence.

22. Page 4, line 8: So, did you apply MPR for the current study? This is not really clear

for me.

Yes, we applied MPR in the mHM simulation. We also mentioned it in the revised manuscript at page

23. Figure 1: I cannot find the explanations for GOCAD, GO20GS and PEST in the text/paragraph.

The original Figure 1 includes many external softwares that are dynamically linked to mHM#OGS. These external softwares are not the core of manuscript, so we modified Figure 1 to avoid misunderstanding. For the new figure about the coupling schematic, please check Figure 2 in the revised manuscript.

24. Page 5, line GIS2FEM: What does GIS2FEM stand for?

The coupling interface GIS2FEM is used to interpolate and transfer mHM grid-based fluxes to OGS nodal flux values. After reading a raster file of mHM generated fluxes, the interface GIS2FEM interpolates the flux value to the top surface elements of the OGS mesh. For each surface element, if its centroid is within the range of mHM grid cell, the flux of this grid cell is assigned to the corresponding surface element in OGS mesh. After all top surface elements being processed, GIS2FEM will take the face integration calculation, by which the recharge data and baseflow are converted into nodal source terms and assigned to the corresponding OGS mesh nodes.

Please check page 8, line 16-21 of the revised manuscript.

25. Page 6, lines 7-15: Could you please check this part. I guess that there are some missing lines or sentence here. For example, I cannot find the introduction and explanation for Eq. 2.

Thank you. We have thoroughly restructured the equations used in the manuscript. Please see the Section 2.1 and 2.2 of the revised manuscript.

26. Page 7, lines 9-10: Due to this liner reservoir conceptualization, I guess that the current coupled model mHM#OGS cannot simulate infiltration from surface water bodies (rivers) to groundwater?

Yes, you are right. The linear reservoir conceptualization means that the baseflow is from groundwater to the water bodies. We have elaborated this point in the revised manuscript. Please see page 8, line 3-10.

27. Page 9, line 24: What is VTU?

The VTU (equivalent to VTK) data format is the data format for an open-source, freely available software system **Visualization Toolkit** (**VTK**), which is used for 3D computer graphics, image processing and visualization.

28. Section 2.5: Please rewrite this section, particularly to clarify/confirm the following: - So, you have two scenarios of groundwater modelling: SC1: spatially distributed recharge and SC2: homogeneous recharge - Did you calibrate both scenarios groundwater modelling independently? Or, did you just calibrate SC1 and then using the calibrated SC1 parameters for SC2?

Thank you for this important question. We do groundwater modeling following a two-step procedure. First, we calibrate the steady state groundwater model against the long term mean of groundwater heads. For this step, we calibrate the model separately in SC1 (renamed as mR in the revised manuscript) and SC2 (renamed as RR in the revised manuscript) so that the K values were adjusted to fit the observations. The second step is to run the transient groundwater model using specific yield and specific storage values according to the literature. The K values in SC1 and SC2 are therefore different. We did not calibrate the transient groundwater model. Instead, we performed the sensitivity analysis of recharge scenarios (i.e., mHM generated recharge vs. homogeneous recharge) using the same storage parameters.

Please check the page 12, line 30-31and page 13, line 1-5 of the revised manuscript.

29. Page 16, line 2: Please provide the unit (m2?) for 8625 and 464.74.

We have followed the reviewer's advice and modified this sentence accordingly.

Please check page 13, line 19 of the revised manuscript.

30. Page 16, lines 2-3: What do you mean by the calibration is robust with totally 114 model runs?

It means the objective function in the calibration was successfully converged after a limited number of model runs, which demonstrated the inverse process is well posed.

31. Page 16, lines 3-4: What do you mean by ' convergence criteria relevant to observation'? Please rephrase the sentence.

Thank you. We have rephrased this sentence in the revised manuscript.

32. Page 16, lines 14-16: The sentence does not flow with the previous ones.

We have followed the reviewer's advice and modified this sentence accordingly.

33. Figure 9: I guess this map is for a steady-state condition. Please clarify.

Yes, it is steady-state calibration. We have followed the reviewer's advice and modified this sentence accordingly.

34. Page 16, line 19-20: What do you mean by the last sentence, i.e. the coincidence with Wechsung (2005)? Is it possible to include/visualize some figures from Wechsung (2005)?

Thank you. Wechsung (2005) depicted the regionalized observations of groundwater head in Naegelstedt catchment. The simulated groundwater head depth (Figure 9) shows a good match with the regionalized observations in Wechsung (2005). Unfortunately due to the copyright issue, we cannot include the figure in the manuscript.

35. Figure 10: Could you please also provide other performance metrics, e.g. NSE, KGE? I missed some crucial information, such as the resolution of the forcing data used and the resolution of mHM model used.

Thank you for the insightful comment. We have followed the reviewer's suggestion and include NSE (0.88) in the updated figure (see Figure 6 in the revised manuscript).

The point data at weather stations were subsequently krigged into a 4 km precipitation fields, and then downscaled to mHM grid cells. The resolution of mHM grid cell is 500m.

36. Page 19, line 5: : : : each monitoring well ... (singular)
Page 19, lines 15-17, Page 20: Please check the English. An example: Another reason is that we assigned a homogenous storage coefficient (?) in all aquifers, which an oversimplified setting.

Thank you for this comment. We have followed the reviewer's advices and made the corresponding change. We also carefully checked the English syntax according to the reviewer's insights.

37. Page 21, line 14: Did Kumar et al. (2016) also simulate groundwater heads?

In the study of Kumar et al., 2016, Kumar et al. used the distributed ground head observations in southern Germany and the central Netherlands to reveal the strong spatial variability of groundwater head fluctuations. His study is based on spatially distributed head observations at large scale. Although he did not simulate the groundwater head, his study can still be used as a proof of the strong spatial variability of groundwater heads.

38. Page 22, lines 9-17: For prediction/application in ungauged basins, I believe that hydrogeological characterization (in ungauged basins) still remains as one of the

main challenges.

This is a very insightful comment. We agree with the reviewer that the comprehensive understanding and characterization of hydrogeological processes in ungauged basins is very challenging and remains an open topic. Therefore, we deleted the sentences related to ungauged basin. Please check page 18, line 10-14 in the revised manuscript.

[revised manuscript text omitted]

---

## Referee Report (RR1)

**Improved groundwater representation at regional scale ( – coupling of mesoscale Hydrologic Model (mHM )

[referee-annotated manuscript omitted]

---

## Author Response (AR2)

**Response to the referee reveiew by Prof.Dr. Cirpka**

**February 2018**

We thank Prof.Dr. Cirpka very much for his efforts devoted in reviewing our manuscript. His deep thoughts and constructive comments really benefit us a lot. We have revised the manuscript carefully according to the two reviews. We replied (in black) every single question (in blue) raised by the reviewers and itemized them in below.

**1 Major Remarks**

1. The English of the paper needs severe polishing. I have attached a corrected version, suggesting chances in almost every sentence. I highly recommend that the authors address a native speaker who surely must be available in an institution the size of UFZ before submitting a revision.

   Response: We thank you for your efforts in correcting our language mistakes. We also checked the English carefully with native English speakers. Please feel free to check the revised manuscript.

2. Unfortunately, I must say that the coupling scheme is not described very clearly. In other schemes, like the USGS-based modeling framework GS-FLOW, a conceptual hydrological model is used to represent surface and soil processes, while the entire groundwater component is handled by the Darcy-based groundwater model. This is not what the authors do. The conceptual hydrological model mHm has its own groundwater storage (denote $x_6$) that remains untouched by the coupling. While the authors describe to some extent how mHm-based groundwater recharge is transferred to nodal loads in the groundwater model, it is not clear how groundwater leaves the OpenGeoSys domain. The authors somehow mention that OpenGeoSys and mHm have different representations of streams so that I somehow speculate that the OpenGeoSys model assumes fixed hydraulic heads along rivers, but it is not explained how river stages are calculated within or for OpenGeoSys. And it is also not clear whether any comparisons between the baseflow computed by mHm and the groundwater flux leaving the OpenGeoSys domain via river nodes are performed.

Response: These comments are very important for us to improve the description of coupling scheme. As pointed out by the reviewer, the key difference between our coupling methods and the others, is that we use OpenGeoSys as a post-processor to realize groundwater head, meanwhile we use mHM to determine component-wise water budget. The reviewer raised two important questions: (1) How does OGS calculate the river stages? (2) How to make sure the river discharge from groundwater calculated by OGS is the same with that from mHM?

For the first question, we simply do not calculate the river stage explicitly. Most coupled surface-subsurface hydrological models characterize river-groundwater interaction based on either first-order flow exchange or boundary condition switching [2]. These methods inevitably rely on a parameter set describing geometric, topographic, and hydraulic properties of the stream channel (e.g., river bed conductance, river bed and drain elevations, channel width). Unfortunately, these parameters are essentially unknown at large scale due to the lack of data and the subgrid-scale topography. Due to these limitations, we use an alternative approach which is based on the step-wise routed baseflow estimated by mHM.

Streams in mHM are implicitly defined based on pre-processing of DEM data and a routing scheme, while OGS uses an explicit predefined river geometry. In OGS, each reach of the stream network is defined by a polyline in a geometry file. To coordinate the two different conceptualizations, we developed a model interface RIV2FEM to convert the routed baseflow estimated by mHM to Neumann boundary conditions assigned at stream nodes at the OGS mesh. The illustration of RIV2FEM scheme can be found in Figure 2 of the revised manuscript. Specifically, the step-wise routed baseflow estimated by mHM is transferred to the uniformly disaggregated discharges by distributing it uniformly along the predefined stream network in OGS:

$$\overline{q}_4(t) = \frac{Q_4(t)}{N} \tag{1}$$

where $\overline{q}_4(t)$ denotes the disaggregated discharge assigned at every stream node in OGS at time $t$ [L$^3$T$^{-1}$], $Q_4(t)$ denotes the routed baseflow at the outlet of catchment at time $t$ [L$^3$T$^{-1}$], $N$ denotes the total number of stream nodes in OGS. The uniformly disaggregated discharges are then assigned to every stream node in OGS to serve as Neumann boundary condition. This approach significantly reduces the number of parameters, avoids the uncertainty caused by the unknown river properties, and is suitable for many real-world applications with scarce data. Moreover, as recharge and baseflow are directly taken from mHM, the mass conservation criteria is naturally satisfied in this approach.

The above statements are also included in the revised manuscript (please refer to P8, L28 - P9, L8).

3. I see the advantages that the authors want to gain by their scheme, but

*I do not understand why they don't explicitly write what they are really doing and why. My understanding is that OpenGeoSys is mainly used as a postprocessor of mHm to obtain groundwater tables, whereas the water budget is calculated by mHm alone. The advantage is pretty clear as the entire stochastic calibration procedure of mHm remains intact and the quality of discharge calculations is not hampered by using a computationally expensive groundwater model for baseflow that requires spatially distributed parameters, which are difficult to obtain. Of course, the Darcy-based groundwater model needs to be calibrated to match heads, but that does not affect the water balance. I assess that using the groundwater model as postprocessor has its intellectual elegance, but then I suggest selling the scheme this way. Otherwise the authors run into the problem why they double-account for groundwater in their two model components.*

Response: Thank you for your insightful comments. We totally agree with the reviewer on the description of "OpenGeoSys is mainly used as a postprocessor of mHm to obtain groundwater tables, whereas the water budget is calculated by mHm alone". This is a good summary of our coupling approach. In order to better convey what we are doing and why, we modified the paragraph describing the modeling motivation as follows:

"The coupling initiative aims to add additional predictive capability of groundwater heads, which is achieved by OGS, to the existing predictive capability of discharge achieved by mHM. On one hand, mHM is used to estimate step-wise and component-wise water budget through model calibration against discharge. On the other hand, OGS serves as a post-processor of mHM to obtain groundwater heads by using driving forces obtained from mHM. Two model interfaces, namely GIS2FEM and RIV2FEM, have been developed to link the two models by transferring recharge and baseflow from mHM to Neumann boundary conditions in OGS."

The above statements are also included in the revised manuscript (please refer to P8, L3-10).

4. *It is also clear that the presented coupled framework cannot overcome difficulties of mHm, or any other similar model, in representing feedbacks of shallow groundwater tables on evapotranspiration, if the shallowness of the groundwater table is caused by lateral groundwater flows. This would require two-way coupling, replacing the groundwater storage of mHm by the Darcy simulator, and joint calibration. Whether the authors will ever implement the two-way coupling, as announced in the outlook of the paper, is doubtful as the modeling and calibration philosophies of the two model components are not particularly compatible.*

Response: We agree with the reviewer that our current coupled model can not overcome the difficulties of typical bucket-type hydrological models. This limitation has been stated in the manuscript (Page 23, Line 25-28): "The main limitation of one-way coupling is that the effects of a shallow

depth to groundwater on actual ET, maintained by lateral groundwater flow, cannot be explicitly addressed."

In terms of the possibility of a two-way coupling, we have observed a successful example, in which a bucket-type hydrological model PCR-GLOBWB is fully coupled to the groundwater simulator MODFLOW [1]. I agree with the reviewer that the two-way coupling is super challenging because of the distinct model concepts. mHM is a "top-down" model which has a strong predictive capability but poor interpretability while OGS is a "bottom-up" model. We still believe that it is feasible to realize the two-way coupling in the future, although it requires restructuring mHM source code. We stated this point in a conservative way in the manuscript:" However, in a future work, we will devote to incorporate a full, two-way coupling using the next version of mHM#OGS model." (P22, L28-29)

5. I am not quite happing with the scheme used to map mHm-based groundwater recharge to OpenGeoSys. In the example application, it seems to be no problem, as apparently the groundwater model simply uses a rectangular grid with finer resolution. (At least the authors don't state otherwise.) In the more general case of mapping gridded fluxes to an unstructured FEM grid, checking for element centroids could be dangerous, and would definitely not ne consistent with the FEM formulation. The consistent formulation would use the weighting functions of the FEM scheme:

$$Q_i^{in} = -\int_{\partial\Omega} W_i(\mathbf{x}) q_{mHm}^{ex}(\mathbf{x}) d(\mathbf{x})$$

in which $Q_i^{in}$ is the nodal load of node $i$, $\partial\Omega$ is the surface boundary of the FEM domain, $W_i$ is the weighting function of node $i$, $\mathbf{x}$ is the spatial coordinate on the surface, and $q_{mHm}^{ex}$ is the exchange rate provided by mHm.

Response: Thank you for your insightful comments. However, the mapping method in GIS2FEM is exactly the same one that the reviewer raised. In the manuscript, we stated that GIS2FEM checks the centroid of element, which is not true and will mislead the readers. We apologize for this unclear description. Actually, we checked the nodes in each elements. GIS2FEM searches for the mHM grid cell that the node is located in, and assigns the recharge value of this grid cell to the corresponding node (marked as $C^m$). After all top surface elements have been processed, GIS2FEM takes subsequently the face integration calculation, by which the specific recharge $C^m$ [LT$^{-1}$] calculated by mHM is converted into volumetric recharge $C^{in}$ [L$^3$T$^{-1}$] and assigned to the corresponding OGS mesh nodes. Specifically, the specific recharge $C$ in a certain element is calculated as:

$$C(\mathbf{x}) = \sum_{j=1}^{N} W_j(\mathbf{x}) C_j^m, \tag{2}$$

where $\mathbf{x}$ is the spatial coordinate on the surface, $N$ is the total number of nodes in a surface element, $W_j$ is the weighting function of node $j$, $C_j^m$ is the specific recharge at node $j$ calculated by mHM [LT$^{-1}$]. Then the volumetric recharge $C_i^{in}$ at node $i$ ($i$ is the global node index) is calculated by the face integration calculation:

$$C_i^{in} = - \int_{\partial\Omega} W_i(\mathbf{x}) C(\mathbf{x}) d(\mathbf{x}), \tag{3}$$

where $C_i^{in}$ is the volumetric recharge of node $i$ [L$^3$T$^{-1}$], $\partial\Omega$ is the surface boundary of the FEM domain, $W_i$ is the weighting function of node $i$. This method is clearly shown in line 10434-10510 in file rf_pcs.cpp at `https://github.com/UFZ-MJ/OGS_mHM/blob/master/FEM/rf_pcs.cpp`

The above statements are also included in the revised manuscript (please refer to P9, L13-27).

6. How does OpenGeoSys calculate river stages? This is completely missing, and knowing discharge along the streams is not sufficient for the groundwater model.

Response: Thank you for your question. We did not calculate river stages. Please find the detailed description of boundary conditions in rivers in the response to question 2.

7. The example application takes place in a karstified aquifer. Muschelkalk is a limestone formation with all features of a karst system. This must not remain unmentioned, as the calibrated hydraulic-conductivity values only hold under the assumption of an equivalent porous medium. Of course, the karst features also hamper the calculation of travel times.

Response: Thank you again for this important information. A research by Kohlhepp et al.(2017) found that the karstification only occurs at the base of the upper Muschelkalk formation [3]: "Although it is of a karst-fracture type with partially solution enlarged fractures, the karstification and the development of conduits are limited and concentrated at the formations' very base"[3]. He concluded that karstification is not a dominant factor to the permeability as it is limited at the base of upper Muschelkalk [3].

Nevertheless, we agree that the treatment of karst formation should be mentioned in the manuscript. Accordingly, we have modified the manuscript as follows:"The mo formation has been widely considered as a karstified formation. In this study area, a recent research by [3] has revealed that in the Hainich critical zone, the intense karstification and the conduit are limited at the base of the mo formation. Accordingly, we use the equivalent porous medium approach to characterize the insignificant karst formation. "(P11, L31 - P12, L3).

8. I am very astonished that the Lower Keuper (" Lettenkeuper ") has such high conductivity values given the fact that this formation contains thick clay layers and acts as an aquitard.

Response: Thank you for the question. We use the automatic-calibration code PEST to calibrate the steady state groundwater model by setting a reasonable range of adjustable parameters (in this case, hydraulic conductivity). We set the upper and lower bound of parameters on the basis of the literature [4]. We found that there is a certain degree of parameter uncertainty, which means a range of possible values are compatible with the prediction. Since the Lower Keuper contains confining clay layers that often acts as aquitard, we re-calibrated the model by setting a narrow range of ku. The updated hydraulic conductivity of ku is 2.848e-5. We use the updated parameter values in the revised manuscript. Please check the updated Table 1 (P17) in the revised manuscript.

9. Assuming no-flow boundary-conditions all around the model domain is not reasonable. There must be a lateral groundwater flux leaving the domain in the lowland-part of the aquifer in the Muschelkalk aquifer.

Response: Thank you for your insights. The outer boundary is generally treated as no-flow bouindary except for the northwestern and northeastern edges. We set Dirichlet boundary conditions at the northwestern and northeastern edges on the basis of Sommer et al. [4]. Accordingly, we modified the description of boundary conditions in the manuscript as follows:

"In general, no-flow boundaries are set at the outer perimeters surrounding the basin as well as at the lower aquitard. On the basis of the measurements, a Dirichlet boundary condition is assumed at the northwestern and northeastern edges." (P12, L6-8)

I agree with you that there must be a lateral groundwater flux leaving the domain in the lowland area in the Muschelkalk aquifer. Several studies have revealed that this portion of water is small compared to the portion of baseflow in the study area.

Based on Toth [6], groundwater flow system in a catchment can be divided into local system, intermediate system and regional system. Our study revealed that the dominating flow systems in the study area are local and intermediate systems. The regional system in which the recharge area occupies the water divide and discharge area lies at the bottom of the basin, is negligible. The reason is that the ratio of the depth to impermeable lower boundary to the half-width of the catchment is small (about 0.008) and the general relief of the catchment is small (about 0.01), which means regional flow is almost impossible [6]. In the local and intermediate flow systems in lowland area, the outflow from Muschelkalk formations is negligible compared to the amount of baseflow. The reason is that first, Wechsung [4] and Seidel [5] have demonstrated that the Muschelkalk formations in lowland have significantly lower permeabilities than those in upland. This observation has also been included in our manuscript in Page 11, Line 28-32. The Muschelkalk in lowland acts as aquitard which partly blocks the movement of groundwater from upland to the outlet of

the catchment. Second, the head gradients are directed from southwestern (or southeastern) uplands to the central lowlands. The flow direction is parallel to the direction of bottom of the catchment, which indicates only a small portion of water can flow out at the bottom. Based on above proofs, we can reasonably assume that the outflow in Muschelkalk formations in the vicinity of catchment outlet is less important, and therefore negligible.

**2   Minor Remarks**

1. See my very extended list of remarks (621 comments, not all of them editorial in nature) in the attachment.

   Response: Thank you again for your detailed review of our manuscript and the insightful comments you raised. We revised the manuscript thoroughly according to each items in the extended list of remarks.

2. I would not use an incomprehensible acronym in the title. My suggestion is: Improved groundwater representation at regional scale by coupling of the mesosclale Hydrological model (mHm) to the groundwater model OpenGeoSys (OGS)

   Response: Thank you very much for this kind suggestion. We have changed the title to "Improved regional scale groundwater representation by the coupling of the mesoscale Hydrologic Model (mHM v5.7) to the groundwater model OpenGeoSys (OGS)".

3. I would not report on version numbers in the abstract.

   Response: We have been informed that the journal GMD requires the specific version number:

   "If the model development relates to a single model then the model name and the version number must be included in the title of the paper. If the main intention of an article is to make a general (i.e. model independent) statement about the usefulness of a new development, but the usefulness is shown with the help of one specific model, the model name and version number must be stated in the title. The title could have a form such as, "Title outlining amazing generic advance: a case study with Model XXX (version Y)"."

4. I know that OpenGeoSys can be used to compute saturate-unsaturated flow, solute transport, heat transport, and geomechanics. However, in the given context, the THMC capabilities are reduced to groundwater hydraulics alone.

   Response: We agree that the THMC capability of OGS does not show up in the manuscript. In the field of hydrogeology, there are nevertheless some important multi-physical processes such as density-dependent flow and thermo-haline flow. OGS is suitable in addressing these problems, and can be easily plugged in current coupling framework in the manuscript.

Nevertheless, we followed the reviewer's suggestion and modified the statement of THMC coupling as follows: "Additionally, OGS demonstrates its capability in addressing thermo-hydro-mechanical-chemical (THMC) coupling processes in large-scale hydrologic cycles (not reflected in this study)" (P22, L4-5).

5. Well identification numbers made of ten digits are totally incomprehensible.

Response: Thank you. We have removed all those long-digit numbers. Instead, we use numbers with only two digits. Please check Figure 2, 10, and 11 in the revised manuscript.

**Response to comments raised by Edwin Sutanudjaja**

We thank Dr. Sutanudjaja very much for his excellent reviews. Below you could find the original comments (marked in blue color) and point-by-point response (marked in black color).

I appreciate that the authors have carefully addressed all the concerns I raised in my review. The writing has been improved. Yet, I still have some minor comments and questions that should be considered:

Page 2 Line 17 (P2L17): … Finally …

P2L24-30: This part, starting from the sentence "For example, models ...", does not flow with the previous sentences. Please consider to rephrase it, Here you may want to consider to discuss several limitations in groundwater model packages (e.g. river, evapotranspiration packages in MODFLOW) for simulating surface water and unsaturated zones.

Response: We have modified the manuscript accordingly. Please check P2, L23-29 in the revised manuscript.

P3L7: What's wrong with "kinematic wave approximation" for surface flow process? Could you please elaborate? And what did you mean by "1D riil flow"?

Response: The kinematic-wave approximation neglects the dynamic components of flow that are represented by the derivative terms in the more complete form of the momentum equation referred to as the Saint-Venant equations. 1D rill flow means the surface flow problem is simplified from a full 3-D problem to 1-D problem via some assumptions.

P3L27-28: "The process uncertainty decreases as one goes depper into the subsurface storage. In the subsurface storage, hydrological process are relatively well understood …" I am not sure that I can agree with this statement. Please include some references to support this statement.

Response: The process uncertainty in subsurface hydrology has been well demonstrated by the theory of porous media flow [1].

P4L2: "OGS has been demonstrated its ability of dealing with data uncertainty in groundwater aquifers." Please give references.

Reference: We have added a reference accordingly.

Figure 1: What is K that leaves donward from the groundwater part? How do you include it in your coupling strategy?

Reference: the K in Figure 1 represents the original scheme characterizing Karst process in mHM. This feature of mHM is irrelevant to our study. In the case study, we do not consider the Karst process in mHM, thus K is always zero.

P6L22, Equation 22: For clarity, could you please expand the source/sink term qs? I believe that this should consist at least, the recharge C and the baseflow q4.

Section 2.3: While explaining the coupling mechanism, I suggest to include the variable symbols as given in Sections 2.1 and 2.2. An example is given as follows:

Response: thank you for your comments. We have modified the manuscript accordingly.

P7L4: The basic idea is to feed fluxes generated by mHM, e.g. distributed groundwater recharge C and baseflow q4 ...

P8L14-21, step 3: Please clarify that your OGS mesh actually has a regular shape: square 250 m x 250 m (not an irregular ones as suggested in Fig. 2c).

Response: thank you for your comments. We have modified the manuscript accordingly.

P9L1: Did you consider groundwater abstraction in your modelling experiment? Please clarify.

Response: in this case study, we do not consider the groundwater abstraction. Please refer to Section 3.4.

P9L7: So, your OGS mesh is rectangular grid: 250 m x 250 m? (not an irregular ones as suggested in Fig. 2c).

Response: yes, we use a structural 3-D mesh in this study. The resolution of mHM grid cells is 500 m $\times$ 500 m. OGS uses a structured, hexahedral 3-D mesh, with a spatial resolution of 250 m $\times$ 250 m in horizontal direction and 10 m in vertical direction over the whole domain. Please refer to P11, L3-6.

P10, Section 3.2: Please include the number of aquifer layers in the OGS model. 11?

Response:  thank you for your comments. We have modified the manuscript accordingly.

P11, Section 3.3: Please clarify to which OGS aquifer layer you force the recharge and baseflow. To the uppermost one only?
Response: yes, we force the recharge and baseflow to the uppermost layer.

P11L21: 1500 mm/month is a very high threshold. Please check.

Response: thank you for pointing out this error. We have modified the statement to: " Only streams with a runoff rate higher than the threshold (in this case study, 0.145 m3/s) are delineated as valid streams." Please refer to P13, L1-3 in the revised manuscript.

P12L23: What are K values? Hydraulic conductivities? Or K values in Fig. 1? Please clarify.

P13L26: What is K? The calibrated K values?

Response: thank you for this suggestion. For sake of consistency, we have removed all the expression of "K values". Instead, we use the alternative expression of "hydraulic conductivities" throughout the manuscript.

P14L8, Figure 8: Please use the same maximum and minimum values on the color scales for all sub-figures in Fig. 8 (page 15) so that they can easily be compared among each other.

Response: we totally agree with this comment. We have re-generated Figure 9 and Figure 10 using the same color scheme with Figure 11. Please refer to the revised manuscript.

**Reference**:
[1] Darcy, Henry. *Les fontaines publiques de la ville de Dijon: exposition et application...* Victor Dalmont, 1856.

[revised manuscript text omitted]